# MULTI-SCALE REPRESENTATIONS BY VARYING WINDOW ATTENTION FOR SEMANTIC SEGMENTATION

**Haotian Yan,**[*] **Ming Wu**[†] **& Chuang Zhang**[‡]
Artificial Intelligence School
Beijing University of Posts and Telecommunications, China
`{yanhaotian,wuming,zhangchuang}@bupt.edu.cn`

## ABSTRACT

Multi-scale learning is central to semantic segmentation. We visualize the effective receptive field (ERF) of canonical multi-scale representations and point out two risks learning them: *scale inadequacy* and *field inactivation*. A novel multi-scale learner, **varying window attention** (VWA), is presented to address these issues. VWA leverages the local window attention (LWA) and disentangles LWA into the query window and context window, allowing the context's scale to vary for the query to learn representations at multiple scales. However, varying the context to large-scale windows (enlarging ratio $R$) can significantly increase the memory footprint and computation cost ($R^2$ times larger than LWA). We propose a simple but professional re-scaling strategy to zero the extra induced cost without compromising performance. Consequently, VWA uses the same cost as LWA to overcome the receptive limitation of the local window. Furthermore, depending on VWA and employing various MLPs, we introduce a multi-scale decoder (MSD), **VWFormer**, to improve multi-scale representations for semantic segmentation. VWFormer achieves efficiency competitive with the most compute-friendly MSDs, like FPN and MLP decoder, but performs much better than any MSDs. For instance, using nearly half of UPerNet's computation, VWFormer outperforms it by $1.0\% - 2.5\%$ mIoU on ADE20K. At little extra overhead, $\sim 10$G FLOPs, Mask2Former armed with VWFormer improves by $1.0\% - 1.3\%$.

## 1 INTRODUCTION

In semantic segmentation, there are two typical paradigms for learning multi-scale representations. The first involves applying filters with receptive-field-variable kernels, classic techniques like atrous convolution (Chen et al., 2018) or adaptive pooling (Zhao et al., 2017). By adjusting hyper-parameters, such as dilation rates and pooling output sizes, the network can vary the receptive field to learn representations at multiple scales.

The second leverages hierarchical backbones Xie et al. (2021); Liu et al. (2021; 2022) to learn multi-scale representations. Typical hierarchical backbones are usually divided into four different levels, each learning representations on feature maps with different sizes. For semantic segmentation, the multi-scale decoder (MSD) (Xiao et al., 2018; Kirillov et al., 2019; Xie et al., 2021) fuses feature maps from every level (*i.e.* multiple scales) and output an aggregation of multi-scale representations.

Essentially, the second paradigm is analogous to the first in that it can be understood from the perspective of varying receptive fields of filters. As the network deepens and feature map sizes gradually shrink, different stages of the hierarchical backbone have distinct receptive fields. Therefore, when MSDs work for semantic segmentation, they naturally aggregate representations learnt by filters with multiple receptive fields, which characterizes multi-level outputs of the hierarchical backbone.

---

[*]The code and model will be available at `https://github.com/yan-hao-tian/vw`
[†]Contributed Equally to the First Author.
[‡]Corresponding Author.

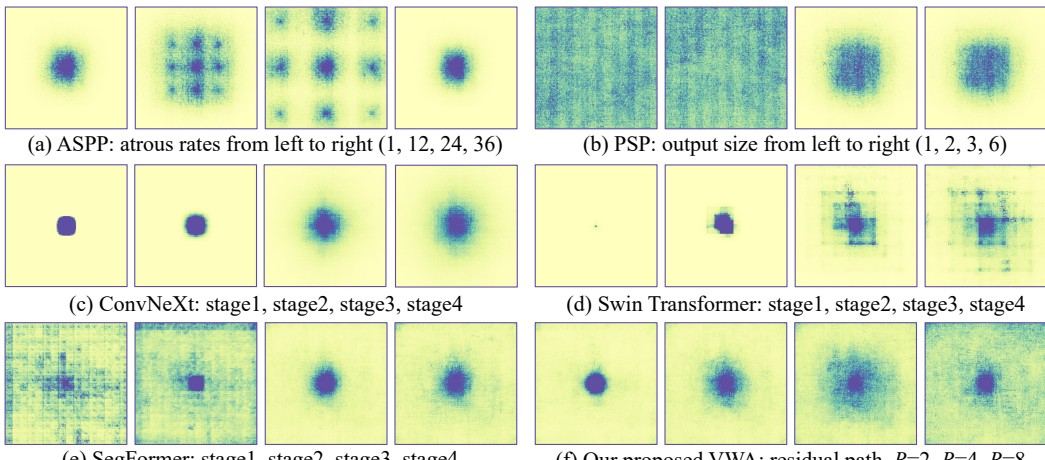

(a) ASPP: atrous rates from left to right (1, 12, 24, 36)  (b) PSP: output size from left to right (1, 2, 3, 6)

(c) ConvNeXt: stage1, stage2, stage3, stage4  (d) Swin Transformer: stage1, stage2, stage3, stage4

(e) SegFormer: stage1, stage2, stage3, stage4  (f) Our proposed VWA: residual path, $R$=2, $R$=4, $R$=8

Figure 1: ERFs of multi-scale representations learned by (a) ASPP, (b) PSP, (c) ConvNeXt, (d) Swin Transformer, (e) SegFormer, and (f) Our proposed **varying window attention**. ERF maps are visualized across 100 images of ADE20K validation set. See Appendix A for more detailed analysis.

To delve into the receptive field of these paradigms, their effective receptive fields (ERF) (Luo et al., 2016) were visualized, as shown in Fig. 1a-e. For the first paradigm, methods like ASPP (applying atrous convolution) Chen et al. (2018) and PSP Zhao et al. (2017) (applying adaptive pooling) were analyzed. For the second paradigm, ERF visualization was performed on multi-level feature maps of ConvNeXt (Liu et al., 2022), Swin Transformer Liu et al. (2021), and SegFormer (MiT) Xie et al. (2021). Based on these visualizations, it can be observed that learning multi-scale representations faces two issues. On the one hand, there is a risk of *scale inadequacy*, such as missing global information (Swin Transformer, ConvNeXt, ASPP), missing local information (PSP), or having only local and global information while missing other scales (SegFormer). On the other hand, there are inactivated areas within the spatial range of the receptive field, as observed in ASPP, Swin Transformer, and the low-level layers of SegFormer. We refer to this as *field inactivation*.

To address these issues, a new way is explored to learn multi-scale representations. This research focuses on exploring *whether the local window attention (LWA) mechanism can be extended to function as a relational filter whose receptive field is variable to meet the scale specification for learning multi-scale representations in semantic segmentation while preserving the efficiency advantages of LWA*. The resulting approach is **varying window attention** (VWA), which learns multi-scale representations with no room for *scale inadequacy* and *field inactivation* (See Fig. 1f). Specifically, VWA disentangles LWA into the query window and context window. The query remains positioned on the local window, while the context is enlarged to cover more surrounding areas, thereby varying the receptive field of the query. Since this enlargement results in a substantial overhead impairing the high efficiency of LWA ($R^2$ times than LWA), we analyze how the extra cost arises and particularly devise pre-scaling principle, densely overlapping patch embedding (DOPE), and copy-shift padding mode (CSP) to eliminate it without compromising performance.

More prominently, tailored to semantic segmentation, we propose a multi-scale decoder (MSD), **VWFormer**, employing VWA and incorporating MLPs with functionalities including multi-layer aggregation and low-level enhancement. To prove the superiority of VWFormer, we evaluate it paired with versatile backbones such as ConvNeXt, Swin Transformer, SegFormer, and compare it with classical MSDs like FPN (Lin et al., 2017), UperNet (Xiao et al., 2018), MLP-decoder (Xie et al., 2021), and deform-attention (Zhu et al., 2020) on datasets including ADE20K (Zhou et al., 2017), Cityscapes (Cordts et al., 2016), and COCOStuff-164k (Caesar et al., 2018). Experiments show that VWFormer consistently leads to performance and efficiency gains. The highest improvements can reach an increase of 2.1% mIoU and a FLOPs reduction of 45%, which are credited to VWA rectifying multi-scale representations of multi-level feature maps at costs of LWA.

In summary, this work has a three-fold contribution:

— We make full use of the ERF technique to visualize the scale of representations learned by existing multi-scale learning paradigms, including receptive-field-variable kernels and different levels of hierarchical backbones, revealing the issues of *scale inadequacy* and *field inactivation*.

— We propose VWA, a relational representation learner, allowing for varying context window sizes toward multiple receptive fields like variable kernels. It is as efficient as LWA due to our pre-scaling principle along with DOPE. We also propose a CSP padding mode specifically for perfecting VWA.

— A novel MSD, VWFormer, designed for semantic segmentation, is presented as the product of VWA. VWFormer shows its effectiveness in improving multi-scale representations of hierarchical backbones, by surpassing existing MSDs in performance and efficiency on classic datasets.

## 2 RELATED WORKS

### 2.1 MULTI-SCALE LEARNER

The multi-scale learner is deemed the paradigm utilizing variable filters to learn multi-scale representations. Sec. 1 has introduced ASPP and PSP. There are also more multi-scale learners proposed previously for semantic segmentation. These works can be categorized into three groups. The first involves using atrous convs, e.g. ASPP, and improving its feature fusion way and efficiency of atrous convolution (Yang et al., 2018; Chen et al., 2018). The second involves extending adaptive pooling, incorporating PSP into other types of representation learners (He et al., 2019a) (He et al., 2019b). However, there are issues of *scale inadequacy* and *field inactivation* associated with these methods' core mechanisms, *i.e.* atrous convs and adaptive pooling, as analyzed in Sec. 1.

The third uses a similar idea to ours, computing the attention matrices between the query and contexts with different scales, to learn multi-scale representations in a relational way for semantic segmentation or even image recognition. In the case of Yuan et al. (2018) and Yu et al. (2021), their core mechanisms are almost identical. As for Zhu et al. (2019), Yang et al. (2021), and Ren et al. (2022), the differences among the three are also trivial. We briefly introduce Yuan et al. (2018) and Zhu et al. (2019), visualizing their ERFs and analyzing their issues (See Fig. 7 and Appendix B for more information). In a word, all of the existing multi-scale learners in a relational way (also known as multi-scale attention) do not address the issues we find, *i.e. scale inadequacy* and *field inactivation*.

### 2.2 MULTI-SCALE DECODER

The multi-scale decoder (MSD) fuses multi-scale representations (multi-level feature maps) learned by hierarchical backbones. One of the most representative MSDs is the Feature Pyramid Network (FPN) (Lin et al., 2017), originally designed for object detection. It has also been applied to image segmentation by using its lowest-level output, even in SOTA semantic segmentation methods such as MaskFormer (Cheng et al., 2021). Lin et al. (2017) has also given rise to methods like (Kirillov et al., 2019) and (Huang et al., 2021). In Mask2Former (Cheng et al., 2022), FPN is combined with deformable attention Zhu et al. (2020) to allow relational interaction between different level feature maps, achieving higher results. Apart from FPN and its derivatives, other widely used methods include the UperNet (Xiao et al., 2018) and the lightweight MLP-decoder proposed by SegFormer.

In summary, all of these methods focus on how to fuse multi-scale representations from hierarchical backbones or enable them to interact with each other. However, our analysis points out that there are *scale inadequacy* and *field inactivation* issues with referring to multi-level feature maps of hierarchical backbones as multi-scale representations. VWFormer further learns multi-scale representations with distinct scale variations and regular ERFs, surpassing existing MSDs in terms of performance while consuming the same computational budget as lightweight ones like FPN and MLP-decoder.

## 3 VARYING WINDOW ATTENTION

### 3.1 PRELIMINARY: LOCAL WINDOW ATTENTION

Local window attention (LWA) is an efficient variant of Multi-Head Self-Attention (MHSA), as shown in Fig. 2a. Assuming the input is a 2D feature map denoted as $\mathbf{x}_{2d} \in \mathbb{R}^{C \times H \times W}$, the first step is reshaping it to local windows, which can be formulated by:

$$\hat{\mathbf{x}}_{2d} = \text{Unfold}\left(\text{kernel} = P, \text{stride} = P\right)(\mathbf{x}_{2d}), \tag{1}$$

where $\text{Unfold}()$ is a Pytorch (Paszke et al., 2019) function (See Pytorch official website for more information). Then the MHSA operates only within the local window instead of the whole feature.

To show the efficiency of local window attention, we list its computation cost to compare with that of MHSA on the global feature (**G**lobal **A**ttention):

$$\Omega\left(\text{GA}\right) = 4(HW)C^2 + 2(HW)^2 C, \qquad \Omega\left(\text{LWA}\right) = 4(HW)C^2 + 2(HW)P^2 C. \quad (2)$$

Note that the first term is on linear mappings, *i.e.*, query, key, value, and out, and the second term is on the attention computation, *i.e.*, calculation of attention matrices and the weighted-summation of value. In the high-dimensional feature space, $P^2$ is smaller than $C$ and much smaller than $HW$. Therefore, the cost of attention computation in LWA is much smaller than the cost of linear mappings which is much smaller than the cost of attention computation in GA.

Besides, the memory footprints of GA and LWA are listed below, showing the hardware-friendliness of LWA. The intermediate outputs of the attention mechanism involve *query*, *key*, *value*, and *out*, all of which are outputs of linear mappings, and attention matrices output from attention computation.

$$\text{Mem.}\left(\text{GA}\right) \propto (HW)C + (HW)^2, \qquad \text{Mem.}\left(\text{LWA}\right) \propto (HW)C + (HW)P^2. \quad (3)$$

The consequence of the computational comparison remains valid. In GA the second term is much larger than the first, but in LWA the second term is smaller than the first.

## 3.2 VARYING THE CONTEXT WINDOW

In LWA, $\hat{\mathbf{x}}_{2\text{d}}$ output by Eq. 1 will attend to itself. In VWA, the query is still $\hat{\mathbf{x}}_{2\text{d}}$, but for the context, by denoting it as $\mathbf{c}_{2\text{d}}$, the generation can be formulated as:

$$\mathbf{c}_{2\text{d}} = \text{Unfold}\left(\text{kernel} = RP, \text{stride} = P, \text{padding} = \text{zero}\right)\left(\mathbf{x}_{2\text{d}}\right), \quad (4)$$

From the view of window sliding, the query generation is a $P \times P$ window with a stride of $P \times P$ sliding on $\mathbf{x}_{2\text{d}}$, and the context generation is a larger $RP \times RP$ window with still a stride of $P \times P$ sliding on $\mathbf{x}_{2\text{d}}$. $R$ is the varying ratio, a constant value in one VWA. As shown in Fig. 2, when $R$ is 1, VWA becomes LWA, and the query and context are entangled together in the local window. But when $R > 1$, with the enlargement of context, the query can see wider than the field of the local window. Thus, VWA is a variant of LWA and LWA is a special case of VWA, where $R = 1$ in VWA.

From the illustration of Fig. 2b, the computation cost of VWA can be computed by:

$$\Omega\left(\text{VWA}\right) = 2\left(R^2 + 1\right)(HW)C^2 + 2(HW)(RP)^2 C. \quad (5)$$

Subtracting Eq. 5 from Eq. 2, the extra computation cost caused by enlarging the context patch is quantified:

$$\Omega\left(\text{EX.}\right) = 2\left(R^2 - 1\right)(HW)C^2 + 2\left(R^2 - 1\right)(HW)P^2 C. \quad (6)$$

For the memory footprint of VWA, it can be computed by according to Fig. 2b:

$$\text{Mem.}\left(\text{VWA}\right) \propto \left(R^2\right)(HW)C + (HW)(RP)^2. \quad (7)$$

Subtracting Eq. 7 from Eq. 3, the extra memory footprint is:

$$\text{Mem.}\left(\text{EX.}\right) \propto \left(R^2 - 1\right)(HW)C + \left(R^2 - 1\right)(HW)P^2. \quad (8)$$

Apparently, the larger the window, the more challenging the problem becomes. First, the efficiency advantage of attention computation (the second term) in LWA does not hold. Second, linear mappings, the first term, yield much more computation budget, which is more challenging because to our knowledge existing works on making attention mechanisms efficient rarely take effort to reduce both the computation cost and memory footprint of linear mappings and their mapping outputs. Next, we will introduce how to address the dilemma caused by varying the context window.

## 3.3 ELIMINATING EXTRA COSTS

With the analysis of Eq. 6 and Eq. 8, the most straightforward way to eliminate the extra cost and memory footprint is re-scaling the large context $\in \mathbb{R}^{C \times R \times P \times R \times P}$ back to the same size as that of the local query $\in \mathbb{R}^{C \times P \times P}$, which means $R$ is set to 1 and thereby both of Eq. 6 and Eq. 8 is 0.

Above all, it is necessary to clarify the difference between using this idea to deal with the extra computation cost and the extra memory footprint. As shown in Fig. 2b, the intermediate produced

Figure 2: (a) illustrates that in LWA, $\mathbf{Q}$, $\mathbf{K}$, and $\mathbf{V}$ are all transformed from the local window. (b) illustrates a naive implementation of VWA. $\mathbf{Q}$ is transformed from the local window. $\mathbf{K}$ and $\mathbf{V}$ are re-scaled from the varing window. PE is short for Patch Embedding. $R$ (of $RP$) denotes the size ratio of the context window to the local window (query). (c) illustrates the professional implementation of VWA. DOPE is short for densely-overlapping Patch Embedding.

by varying (enlarging) the window, which is the output of Eq. 4, already takes the memory that is $R^2(HW)C$. Therefore, re-scaling the large context after generating it does not work, the right step should be re-scaling the feature $\mathbf{x}_{2d}$ before running Eq. 4. We name this pre-scaling principle.

Solving the problem is begun by the pre-scaling principle. A new feature scaling paradigm, densely overlapping patch embedding (DOPE), is proposed. This method is different from patch embedding (PE) widely applied in ViT and HVT as it does not change the spatial dimension but only changes the dimensionality. Specifically, for $\mathbf{x}_{2d}$, after applying Eq. 4 on it, the output's shape is:

$$H/P \times W/P \times RP \times RP \times C. \tag{9}$$

which produces the memory footprint of $R^2 HWC$. Instead, DOPE first reduces the dimensionality of $\mathbf{x}_{2d}$ from $C$ to $C/R^2$, and then applies Eq. 4, resulting in the context with a shape of:

$$H/P \times W/P \times RP \times RP \times C/R^2. \tag{10}$$

which produces the memory footprint of $HWC$, the same as $\mathbf{x}_{2d}$, eliminating the extra memory.

Since PE is often implemented using conv layers, how DOPE re-scales features is expressed as:

$$\text{DOPE} = \text{Conv2d}(\text{in} = C, \text{out} = C/R^2, \text{kernel} = R, \text{stride} = 1). \tag{11}$$

So, the term "densely overlapping" of DOPE describes the densely arranged pattern of convolutional kernels, especially when $R$ is large, filtering every position. The computation cost introduced by DOPE can be computed by:

$$\Omega\,(\text{DOPE}) = R \times R \times C \times C/R^2 \times HW = (HW)C^2. \tag{12}$$

This is equivalent to the computation budget required for just one linear mapping.

However, the context window $\in \mathbb{R}^{RP \times RP \times C/R^2}$ processed by DOPE cannot be attended to by the query window $\in \mathbb{R}^{P \times P \times C}$. We choose PE to downsample the context and increase its dimensionality to a new context window $\in \mathbb{R}^{P \times P \times C}$. The PE function can be formulated as:

$$\text{PE} = \text{Conv2d}(\text{in} = C/R^2, \text{out} = C, \text{kernel} = R, \text{stride} = \text{R}). \tag{13}$$

The computation cost for one context window applying PE is:

$$\Omega\,(\text{PE for one context}) = R \times R \times C/R^2 \times C \times RP/R \times RP/R = P^2 C. \tag{14}$$

For all context windows from DOPE, with a total of $H/P \times W/P$, the computation cost becomes:

$$\Omega\,(\text{PE}) = H/P \times W/P \times \Omega\,(\text{PE for one context}) = (HW)C^2. \tag{15}$$

This is still the same as only one linear mapping.

After applying the re-scaling strategy described, as shown in Fig. 2c, it is clear that the memory footprint of VWA is the same as $\text{Mem.}\,(\text{LWA})$ in Eq. 3, not affected by the context enlargement. The attention computation cost is also the same as $\Omega(\text{LWA})$ in Eq. 2. For DOPE, VWA uses it once, thus adding one linear mapping computation to $\Omega(\text{LWA})$. For PE, VWA uses it twice for mapping the key and value from the DOPE's output, replacing the original key and value mapping. So the computation cost of VWA merely increases 25%—one linear mapping of $(HW)C^2$—than LWA:

$$\Omega\,(\text{VWA}) = (4+1)(HW)C^2 + 2(HW)P^2 C. \tag{16}$$

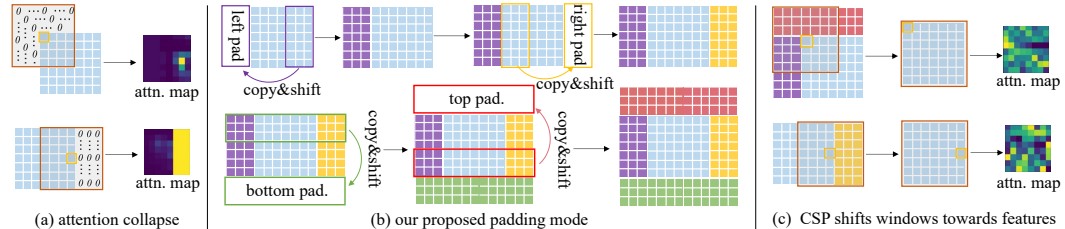

Figure 3: (a) illustrates the zero-padding mode caused attention collapse when the context window is very large and the context window surrounds the local window near the corner or edge. (b) illustrates the proposed copy-shift padding (CSP) mode. The color change indicates where the padding pixels are from. (c) CSP is equivalent to moving the context windows towards the feature, ensuring that every pixel the query attends to has a different valid non-zero value. Best viewed in color.

## 3.4 ATTENTION COLLAPSE AND COPY-SHIFT PADDING

The padding mode in Eq. 4 is zero padding. However, visualizing attention maps of VWA, we find that the attention weights of the context window at the corner and edge tend to have the same value, which makes attention collapse. The reason is that too many same zeros lead to smoothing the probability distribution during Softmax activation. As shown in Fig. 3, to address this problem, we propose copy-shift padding (CSP) equivalent to making the coverage of the large window move towards the feature. Specifically, for the left and right edges, $\mathbf{x}_{2d}$ after CSP is:

$$\mathbf{x}_{2d} = \text{Concat}(d=4)(\mathbf{x}_{2d}[..., (R+1)P/2 : RP], \mathbf{x}_{2d}, \mathbf{x}_{2d}[..., -RP : -(R+1)P/2]). \quad (17)$$

where $\text{Concat}()$ denotes the Pytorch function concatenating a tuple of features along the dimension d. Based on $\mathbf{x}_{2d}$ obtained by Eq. 17, CSP padding the top and bottom sides can be formulated by:

$$\mathbf{x}_{2d} = \text{Concat}(d=3)(\mathbf{x}_{2d}[..., (R+1)P/2 : RP, :], \mathbf{x}_{2d}, \mathbf{x}_{2d}[..., -RP : -(R+1)P/2, :]). \quad (18)$$

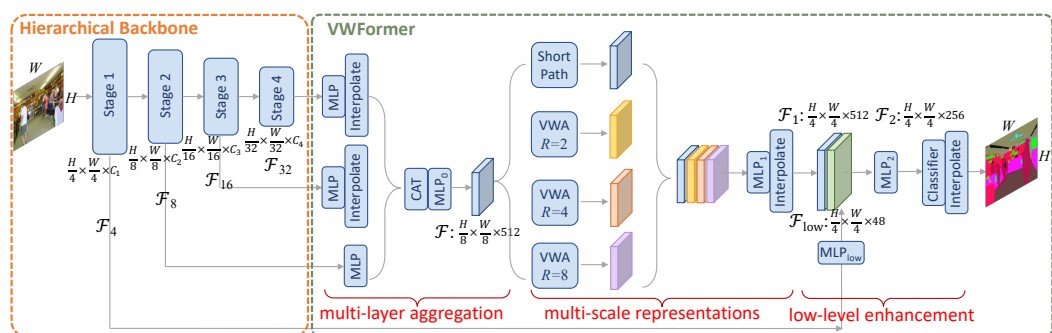

Figure 4: VWFormer contains multi-layer aggregation, learning multi-scale representations, and low-level enhancement. Like other MSDs, VWFormer takes multi-level feature maps as inputs.

## 4 VWFORMER

**Multi-Layer Aggregation**  As illustrated in Fig. 4, VWFormer first concatenates feature maps from the last three stages instead of all four levels for efficiency, by upsampling the last two ($\mathcal{F}_{16}$ and $\mathcal{F}_{32}$) both to the same size as the 2nd-stage one ($\mathcal{F}_8$), and then transform the concatenation with one linear layer ($\text{MLP}_0$) to reduce the channel number, with $\mathcal{F}$ as the outcome.

**Multi-Scale Representations**  To learn multi-scale representations, three VWA mechanisms with varying ratios $R = 2, 4, 8$ are paralleled to act on the multi-layer aggregation's output $\mathcal{F}$. The local window size $P$ of every VWA is set to $\frac{H}{8} \times \frac{W}{8}$, subject to the spatial size of $\mathcal{F}$. Additionally, the short path, exactly a linear mapping layer, consummates the very local scale. The MLPs of VWFormer consist of two layers. The first layer ($\text{MLP}_1$) is a linear reduction of multi-scale representations.

**Low-Level Enhancement**  The second layer ($\text{MLP}_2$) of MLPs empowers the output $\mathcal{F}_1$ of the first layer with low-level enhancement (LLE). LLE first uses a linear layer ($\text{MLP}_{\text{low}}$) with small output channel numbers $48$ to reduce the lowest-level $\mathcal{F}_4$ dimensionality. Then $\mathcal{F}_1$ is upsampled to the same size as $\text{MLP}_{\text{low}}$'s output $\mathcal{F}_{\text{low}}$ and fused with it through $\text{MLP}_2$, outputting $\mathcal{F}_2$.

## 5 EXPERIMENTS

### 5.1 DATASET AND IMPLEMENTATION

Experiments are conducted on three public datasets including Cityscapes, ADE20K, and COCOStuff-164K (See D.2 for more information). The experiment protocols are the same as the compared method's official repository. For ablation studies, we choose the Swin-Base backbone as the testbed and use the same protocols as Swin-UperNet (See D.3 for more information).

### 5.2 MAIN RESULTS

#### 5.2.1 COMPARISON WITH SEGFORMER (MLP-DECODER)

SegFormer uses MixFormer (MiT) as the backbone and designs a lightweight MLP-decoder as MSD to decode multi-scale representations of MixFormer. To demonstrate the effectiveness of VWFormer in improving multi-scale representations by VWA, we replace the MLP-decoder in SegFormer with VWFormer. Table 1 shows the number of parameters, FLOPs, memory footprints, and mIoU. Across all variants of backbone MiT (B0→B5), VWFormer trumps MLP-decoder on every metric.

Table 1: Comparison of SegFormer (MiT-MLP) with VW-SegFormer (MiT-VW.).

| MSD | backbone | ADE20K | | | | Cityscapes | COCO |
| | | params(M) ↓ | FLOPs(G) ↓ | mem.(G)↓ | mIoU(/MS)↑ | mIoU(/MS)↑ | mIoU↑ |
|---|---|---|---|---|---|---|---|
| MLP | MiT-B0 | 3.8 | 8.4 | 2.2 | 37.4 / 38.0 | 76.2 / 78.1 | 35.6 |
| | MiT-B1 | 13.7 | 15.9 | 2.7 | 42.2 / 43.1 | 78.5 / 80.0 | 40.2 |
| | MiT-B2 | 27.5 | 62.4 | 4.3 | 46.5 / 47.5 | 81.0 / 82.2 | 44.6 |
| | MiT-B3 | 47.3 | 79.0 | 5.6 | 49.4 / 50.0 | 81.7 / 83.3 | 45.5 |
| | MiT-B4 | 64.1 | 95.7 | 7.0 | 50.3 / 51.1 | 81.9 / 83.4 | 46.5 |
| | MiT-B5 | 84.7 | 112 | 8.1 | 51.0 / 51.8 | 82.3 / 83.5 | 46.7 |
| VW. | MiT-B0 | 3.7 | 5.8 | 2.1 | 38.9 / 39.6 | 77.2 / 78.7 | 36.2 |
| | MiT-B1 | 13.7 | 13.2 | 2.6 | 43.2 / 44.0 | 79.0 / 80.4 | 41.5 |
| | MiT-B2 | 27.4 | 46.6 | 4.3 | 48.1 / 49.2 | 81.7 / 82.7 | 45.2 |
| | MiT-B3 | 47.3 | 63.3 | 5.6 | 50.3 / 50.9 | 82.4 / 83.6 | 46.8 |
| | MiT-B4 | 64.0 | 79.9 | 7.0 | 50.8 / 51.6 | 82.7 / 84.0 | 47.6 |
| | MiT-B5 | 84.6 | 96.1 | 8.1 | 52.0 / 52.7 | 82.8 / 84.3 | 48.0 |

#### 5.2.2 COMPARISON WITH UPERNET

In recent research, UperNet was often used as MSD to evaluate the proposed vision backbone in semantic segmentation. Before multi-scale fusion, UperNet learns multi-scale representations by utilizing PSPNet (with *scale inadequacy* issue) merely on the highest-level feature map. In contrast, VWFormer can rectify ERFs of every fused multi-level feature map in advance. Table 2 shows VWFormer consistently uses much fewer budgets to achieve higher performance.

Table 2: Comparison of UperNet with VWFormer. Swin Transformer and ConvNeXt serve as backbones. VW-Wide is VWFormer with two times larger channels.

| MSD | backbone | ADE20K | | | | Cityscapes |
| | | params(M) ↓ | FLOPs(G) ↓ | mem.(G)↓ | mIoU(/MS)↑ | mIoU(/MS)↑ |
|---|---|---|---|---|---|---|
| UperNet | Swin-B | 120 | 306 | 8.7 | 50.8 / 52.4 | 82.3 / 82.9 |
| | Swin-L | 232 | 420 | 12.7 | 52.1 / 53.5 | 82.8 / 83.3 |
| | ConvNeXt-B | 121 | 293 | 5.8 | 52.1 / 52.7 | 82.6 / 82.9 |
| | ConvNeXt-L | 233 | 394 | 8.9 | 53.2 / 53.4 | 83.0 / 83.5 |
| | ConvNeXt-XL | 389 | 534 | 12.8 | 53.6 / 54.1 | 83.1 / 83.5 |
| VW. | Swin-B | 95 | 120 | 7.6 | 52.5 / 53.5 | 82.7 / 83.3 |
| | Swin-L | 202 | 236 | 11.5 | 54.2 / 55.6 | 83.2 / 83.9 |
| | ConvNeXt-B | 95 | 107 | 4.6 | 53.3 / 54.1 | 83.2 / 83.9 |
| | ConvNeXt-L | 205 | 208 | 7.7 | 54.3 / 55.1 | 83.5 / 84.1 |
| | ConvNeXt-XL | 357 | 346 | 11.4 | 54.6 / 55.3 | 83.6 / 84.3 |
| VW-Wide | Swin-L | 223 | 306 | 13.7 | 54.7 / 56.0 | 83.5 / 84.2 |

### 5.2.3 COMPARISON WITH MASKFORMER AND MASK2FORMER

MaskFormer and Mask2Former introduce the mask classification mechanism for image segmentation but also rely on MSDs. MaskFormer uses the FPN as MSD, while Mask2Former empowers multi-level feature maps with feature interaction by integrating Deformable Attention (Zhu et al., 2020) into FPN. Table 3 demonstrates that VWFormer is as efficient as FPN and achieves mIoU gains from $0.8\%$ to $1.7\%$. The results also show that VWFormer performs stronger than Deformable Attention with less computation costs. The combo of VWFormer and Deformable Attention further improves mIoU by $0.7\%$-$1.4\%$. This demonstrates VWFormer can still boost the performance of interacted multi-level feature maps via Deformable Attention, highlighting its generability.

Table 3: Comparison of VWFormer with FPN and Deformable Attention. MaskFormer and Mask2Former serve as testbeds (mask classification heads).

| head | MSD | backbone | params(M)↓ | FLOPs(G)↓ | mem.(G)↓ | mIoU(/MS)↑ |
|---|---|---|---|---|---|---|
| MaskFormer | FPN | Swin-T | 41.8 | 57.3 | 4.8 | 46.7 / 48.8 |
| | | Swin-S | 63.1 | 81.1 | 5.5 | 49.4 / 51.0 |
| | | Swin-B | 102 | 126 | 8.2 | 52.7 / 53.9 |
| | | Swin-L | 212 | 239 | 11.5 | 54.1 / 55.6 |
| | VW. | Swin-T | 42.8 | 55.6 | 5.3 | 47.8 / 49.0 |
| | | Swin-S | 64.1 | 79.4 | 6.2 | 50.5 / 52.7 |
| | | Swin-B | 102 | 124 | 8.5 | 53.8 / 54.6 |
| | | Swin-L | 213 | 237 | 12.0 | 55.3 / 56.5 |
| Mask2Former | Deform-Attn | Swin-T | 47.4 | 74.1 | 5.4 | 47.7 / 49.6 |
| | | Swin-S | 68.8 | 97.9 | 6.6 | 51.3 / 52.4 |
| | | Swin-B | 107 | 142 | 9.9 | 53.9 / 55.1 |
| | | Swin-L | 215 | 255 | 13.1 | 56.1 / 57.3 |
| | VW. | Swin-T | 47.7 | 60.8 | 3.6 | 48.3 / 50.5 |
| | | Swin-S | 69.8 | 84.6 | 4.7 | 52.1 / 53.7 |
| | | Swin-B | 108 | 129 | 7.9 | 54.6 / 56.0 |
| | | Swin-L | 217 | 244 | 12.1 | 56.5 / 57.8 |
| | VW. (Deform-Attn) | Swin-T | 53.3 | 85.3 | 6.2 | 48.5 / 50.4 |
| | | Swin-S | 74.7 | 108 | 7.3 | 52.0 / 53.6 |
| | | Swin-B | 113 | 152 | 10.1 | 55.2 / 56.5 |
| | | Swin-L | 221 | 266 | 13.7 | 56.8 / 58.3 |

## 5.3 ABLATION STUDIES

### 5.3.1 SCALE CONTRIBUTION

Table 4 shows the performance drops when removing any VWA of VWFormer. These results indicate every scale is crucial, suggesting that *scale inadequacy* is fatal to multi-scale learning. Also, we add a VWA branch with $R = 1$ context windows which is exactly LWA, and then substitute $R = 2$ VWA with it. The results show LWA is unnecessary in VWFormer because the short path ($1 \times 1$ convolution) in VWFormer can provide a very local receptive field, as visualized in Fig. 1f.

Table 4: Performance of different Scale combinations. Conducted on ADE20K. The numbers of "scale group" are varying ratios. (2, 4, 8) is the default setting.

| backbone | Swin-B | | | | | |
|---|---|---|---|---|---|---|
| scale group | (2, 4, 8) | (2, 4) | (2, 8) | (4, 8) | (1, 2, 4, 8) | (1, 4, 8) |
| mIoU(/MS)↑ | 52.5 / 53.5 | 51.8 / 52.8 | 51.7 / 52.7 | 51.9 / 52.8 | 52.1 / 53.0 | 51.9 / 52.9 |

### 5.3.2 PRE-SCALING VS. POST-SCALING

Table 5 compares: applying VWA without rescaling, with a naive rescaling as depicted in Fig. 2b, and our proposed professional strategy. VWA originally consumes unaffordable FLOPs and memory footprints. Applying the naive scaling strategy saves some FLOPs and memory footprints, but introduces patch embedding (PE) increasing an amount of parameters. Our proposed strategy does not only eliminate the computation and memory introduced by varying the context window but also only adds a small number of parameters. Moreover, it does not sacrifice performance for efficiency.

Table 5: Performance of different ways to re-scale the context window. Conducted on ADE20K.

| backbone | pre. or post. | rescaling method | params.↓ | FLOPs(G)↓ | mem.(G)↓ | mIoU(/MS)↑ |
|----------|---------------|------------------|----------|-----------|----------|------------|
| Swin-B | pre-scaling | DOPE → PE | 95 | 120 | 7.6 | 52.5 / 53.5 |
| | post-scaling | PE | 115 | 210 | 9.9 | 52.3 / 53.6 |
| | post-scaling | Avg. Pooling | 93 | 114 | 9.3 | 51.7 / 52.8 |
| | no rescaling | – | 93 | 315 | 13.1 | 52.4 / 53.4 |

### 5.3.3 ZERO PADDING VS. VW PADDING

The left table of Table 6 shows using zero padding to obtain the context window results in a $0.8\%$ lower mIoU than applying CSP to obtain the context window. Such a performance loss is as severe as removing one scale of VWA, demonstrating the harm of attention collapse and the necessity of our proposed CSP in applying the varying window scheme.

Table 6: **Left:** Performance of zero padding mode and our proposed CSP. **Right:** Performance of different output channel number settings of LLE module in VWFormer.

| backbone | Swin-B | | backbone | Swin-B | | |
|----------|--------|--------|----------|--------|--------|--------|
| padding | zero | CSP | method | — | LLE | FPN |
| mIoU(/MS) | 52.0 / 52.7 | 52.5 / 53.5 | mIoU / FLOPs(G) | 51.8 / 112 | 52.5 / 120 | 52.1 / 176 |

### 5.3.4 EFFECTIVENESS OF LOW-LEVEL ENHANCEMENT

The right table of Table 6 analyzes Low-Level Enhancement (LLE). First, removing LLE degrades mIoU by $0.7\%$. From Fig. 1, it can be seen that the lowest-level feature map is of unique receptivity, very local or global, adding new scales to VWFormer's multi-scale learning. FPN is also evaluated as an alternative, and the results show FPN is neither stronger nor cheaper than LLE.

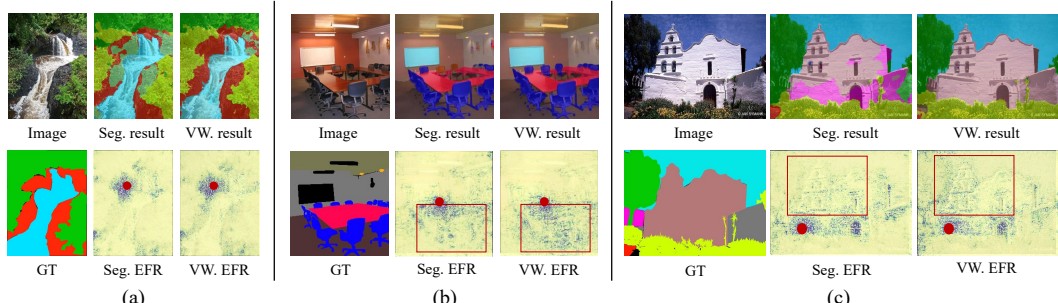

Figure 5: Visualization of inference results and ERFs of SegFormer and VWFormer. The red dot is the query location. The red box exhibits our method's receptive superiority. Zoom in to see details.

## 6 SPECIFIC ERF VISUALIZATION

The ERF visualization of Fig. 1 is averaged on many ADE20k val images. To further substantiate the proposed issue, Fig. 5 analyzes the specific ADE20K val image with ERFs of segformer and VWFormer contrastively. This new visualization can help to understand the receptive issue of existing multi-scale representations and show the strengths of VWFormer's multi-scale learning.

Fig. 5a showcases a waterfall along with rocks. Our VWFormer's result labels most of the rocks, but SegFormer's result struggles to distinguish between "rock" and "mountain". From their ERFs, it can be contrastively revealed that VWFormer helps the query to understand the complex scene, even delineating the waterfall and rocks, more distinctly than SegFormer within the whole image.

Fig. 5b showcases a meeting room with a table surrounded by swivel chairs. Our VWFormer's result labels all of the swivel chairs, but SegFormer's result mistakes two swivel chairs as general chairs. From their ERFs, it can be contrastively revealed when VWFormer infers the location, it incorporates the context of swivel chairs, within the Red box on the opposite side of the table. But SegFormer neglects to learn about that contextual information due to its scale issues.

Fig. 5c showcases a white tall building. Our VWFormer's result labels it correctly, but SegFormer's result mistakes part of the building as the class "house". From their ERFs, it can be contrastively revealed that VWFormer has a clearer receptivity than SegFormer within the Red box which indicates this object is a church-style building.

ACKNOWLEDGEMENT

This work was supported by the National Natural Science Foundation of China (NSFC) under Grant 62076093.

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

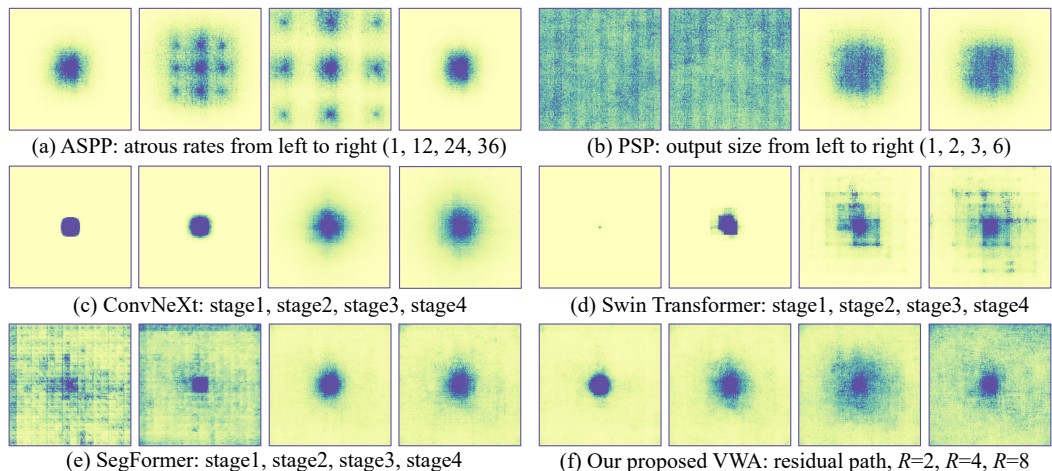

(a) ASPP: atrous rates from left to right (1, 12, 24, 36)  (b) PSP: output size from left to right (1, 2, 3, 6)

(c) ConvNeXt: stage1, stage2, stage3, stage4  (d) Swin Transformer: stage1, stage2, stage3, stage4

(e) SegFormer: stage1, stage2, stage3, stage4  (f) Our proposed VWA: residual path, R=2, R=4, R=8

Figure 6: ERF visualization of multi-scale representations learned by (a) ASPP, (b) PSP, (c) ConvNeXt, (d) Swin Transformer, (e) SegFormer, and (f) Our proposed **varying window attention**. ERF maps are visualized across 100 images of ADE20K validation set. This figure is exactly Fig. 1.

## A  QUALITATIVE ANALYSIS OF TYPICAL METHODS' ERFs

Below is a detailed analysis of the issues with methods visualized in Fig. 1. For good readability, Fig. 1 is copied and pasted here as Fig. 6

**ASPP** employs atrous convs with a set of reasonable fixed atrous rates to learn multi-scale representations. However, as shown in Fig. 6a, the largest receptive field does not capture the desired scale of representations. This is because the parameter settings are manual and do not adapt to the image size. The lack of adaptability becomes more severe when training and testing samples have different sizes, a common occurrence with applying strategies like test-time augmentation (TTA). Furthermore, when the receptive field is large, contributions from the atrous parts are zero, leading to inactivated subareas within larger receptive fields.

**PSP** applies pooling filters with different scales by adjusting the hyper-parameter, output size of adaptive pooling, to learn multi-scale representations. However, as shown in Fig. 6b, the receptive field sizes are exactly the same for output sizes 1 and 2 and for output sizes 3 and 6. This is because the super small output needs to be interpolated to the original feature size. During the interpolation, if a position does not require interpolation to obtain its value, its receptive field remains unchanged. However, if interpolation is needed, the receptive field can be influenced by other positions.

**ConvNeXt** stages' receptive field sizes change from small to large as the network deepens. This is because the stacking of multiple 7x7 convolutions can simulate much larger convolutional kernels. However, as shown in Fig. 6c, compared to ASPP and PSP, the largest receptive field of the four scales in ConvNeXt only covers half of the original image and does not capture a global representation because the 7x7 conv is still of locality. Additionally, it is hard to distinguish between the receptive field scale of the third and the fourth stage.

**Swin Transformer**'s basic layers consist of local window attention mechanisms and shift-window attention mechanisms. The feature maps in its four stages exhibit an increase in receptive field size from small to large. Swin Transformer also faces challenges in learning global representations effectively. Moreover, due to the shift operation of the local window, its receptive field shape is irregular as shown in Fig. 6d, leading to inactivated subareas within the receptive field range.

**SegFormer**'s basic layers are sophisticated, incorporating local window attention, global pooling attention, and 3x3 convolutions. It is hence difficult to imagine the receptive field shape and size for its four-level feature maps. Fig. 6e indicates that SegFormer learns global representations in the low-level layers (i.e., the first and second levels) but still suffers from inactivated subareas within the receptive field range. In the higher layers (i.e., the third and fourth levels), they learn more localized representations but their field ranges are very similar. Therefore SegFormer also meets the *scale inadequacy* because it can only learn global and local representations.

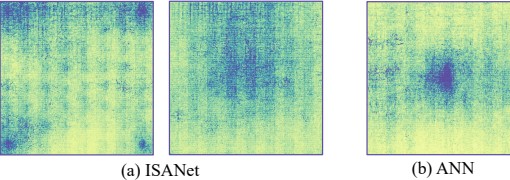

(a) ISANet                                    (b) ANN

Figure 7: (a) contains ERF maps of two-scale (left is global and right is local) representations learnt by ISANet. (b) is the ERF map of global representation learnt by ANN

## B ERFs OF EXISTING MULTI-SCALE ATTENTION (RELATIONAL MULTI-SCALE LEARNER)

Fig. 7a visualizes the ERF map of ISANet (Yuan et al., 2018), merely learning local and global representations while ignoring other scales. So the issue of *scale inadequacy* for ISANet is very clear. The local representation is learned using the local window attention mechanism, while the global representation is obtained by interlacing pixels from all windows to create new windows that contain pixels from each original local window. Then, the window attention mechanism is applied to the new window. The ERF map shows that their receptive fields are not continuous due to interlacing, suggesting that ISANet also meets *field inactivation*.

ANN Zhu et al. (2019) uses adaptive pooling to capture multi-scale features in a PSP manner. Then they are together attended to by the original feature which serves as the query. The scale of the receptive field is singly global because every context filtered by adaptive pooling is derived from the whole feature map. So the issue of *scale inadequacy* is also very clear for ANN. Fig. 7b shows the activation does not spread the global range uniformly and the bottom area is insufficiently activated. Therefore, both *scale inadequacy* and *field inactivation* are issues of ANN and its relevant methods.

The bottom three rows of Table 7 empirically compare ours to ISANet and ANN. VWFormer outperforms both of them by large margins consistently across different backbones and benchmarks.

Table 7: Comparison of VWFormer with other receptive-field-variable multi-scale learners. Red, Green, Blue highlight the top-3 results of one metric.

| | | | ADE20K | | Cityscapes | |
| | | | ResNet101 | ResNet50 | ResNet101 | ResNet50 | ResNet101 |
| Method | Params(M)↓ | FLOPs(G)↓ | mIoU(/MS)↑ | mIoU(/MS)↑ | mIoU(/MS)↑ | mIoU(/MS)↑ |
|---|---|---|---|---|---|---|
| PSPNet | 68.0 | 254.9 | 41.1 / 42.0 | 43.6 / 44.4 | 77.9 / 79.2 | 79.0 / 80.0 |
| DeepLabV3 | 87.1 | 346.1 | 42.4 / 43.3 | 44.0 / 45.2 | 79.3 / 80.7 | 79.7 / 80.8 |
| DeepLabV3+ | 62.6 | 252.8 | 42.7 / 43.7 | 44.6 / 46.0 | 79.9 / 81.0 | 81.0 / 82.2 |
| APCNet | 75.4 | 282.0 | 42.2 / 43.3 | 45.5 / 46.7 | 78.8 / 80.0 | 79.7 / 80.6 |
| DMNet | 72.2 | 273.4 | 42.5 / 43.6 | 45.3 / 46.1 | 79.2 / 80.2 | 79.6 / 80.7 |
| DenseASPP | 96.8 | 461.2 | 42.4 / 43.5 | 43.8 / 44.9 | 79.6 / 80.4 | 80.3 / 81.0 |
| ANN | 65.2 | 262.6 | 41.0 / 42.3 | 43.0 / 44.2 | 78.9 / 80.6 | 78.8 / 80.4 |
| ISANet | 53.7 | 227.9 | 41.1 / 42.4 | 42.6 / 43.1 | 79.3 / 80.5 | 80.6 / 81.6 |
| VWFormer | 50.4 | 203.2 | 43.5 / 44.4 | 45.9 / 47.0 | 80.3 / 81.2 | 81.5 / 82.7 |

## C MORE EXPERIMENTAL ANALYSES

### C.1 COMPARISON OF VWFORMER WITH MULTI-SCALE LEARNERS

To verify the superiority of VWFormer over representative multi-scale learners for semantic segmentation, Table 7 compares VWFormer with PSPNet Zhao et al. (2017), DeepLabV3 Chen et al. (2017), DeepLabV3+ Chen et al. (2018), DenseASPP Yang et al. (2018), APCNet He et al. (2019b), DMNet He et al. (2019a), ANN Zhu et al. (2019), and ISANet Yuan et al. (2018). For fairness, we employ the same backbones for all the methods, including ResNet50 and ResNet101 He et al. (2016). All methods are trained for 80000 iterations, and evaluated on Cityscapes as well as ADE20K. The input size is $768/769 \times 768/769$ for Cityscapes, and $512 \times 512$ for ADE20K.

From Table 7, we can find that VWFormer brings the best results to both ResNet50 and ResNet101 on both datasets. Specifically, on Cityscapes, VWFormer achieves $81.2\%$ mIoU with ResNet50,

and 82.7% mIoU with ResNet101, which are the best results among all methods. DeepLabV3+ achieves the closest performance to ours but has more computation costs and parameters by 49.6G and 12.2M, respectively. On ADE20K, VWFormer outperforms other methods by large margins consistently. APCNet performs most closely to ours, but VWFormer uses the least FLOPs and parameters. In short word, VWFormer is more powerful than any other multi-scale learners.

## C.2 VWFORMER WITH SOTA METHODS

Table 8 analyzes our method briefly with state-of-the-art semantic segmentation methods created on other tracks. **Left** of Table 8 shows the comparison with HRViT Gu et al. (2022), which is a hierarchical Vision Transformer (HVT) with complex multi-scale learning. It was paired originally with MLP-decoder from SegFormer as MSD. Moreover, MLP-decoder is replaced with our VW-Former. The performance gains are considerable, supporting VWFormer's capability of improving multi-scale representations.

**Center** compares VWFormer with SegViT-V2 Zhang et al. (2023). SegViT-V2 is a decoder specifically for ViT (or categorized as plain Vision Transformer). Here VWFormer cooperates with plain Vision Transformer at the first time. The improvement shows that VWFormer is not only effective in HVT but also powerful in plain backbone architecture.

**Right** shows the comparison with ViT-Adapter Chen et al. (2022), which is a pre-training technique for improving ViT on dense prediction tasks. Like many works on Vision Transformer employing UperNet as MSD for semantic segmentation, ViT-Adapter was also originally paired with UperNet. Moreover, UperNet is replaced with VWFormer, achieving considerable performance gains.

Table 8: **Left:** VWFormer paired with HRViT for comparison with original HRViT. **Center:** Comparison of VWFormer with SegViT-V2. **Right:** VWFormer paired with Adapter for comparison with original Adapter (paired with UperNet). Evaluated on ADE20K with multi-scale inference.

| mIoU | HRViT-b1 | b2 | b3 | mIoU | BEiT-V2-L | mIoU | Ada.-B | Ada.-L |
|------|----------|------|------|----------|-----------|------|--------|--------|
| MLP | 45.6 | 48.8 | 50.2 | SegViT-V2 | 58.2 | Uper. | 52.5 | 54.4 |
| VW. | 46.9 | 50.0 | 51.6 | VW. | 58.8 | VW. | 53.5 | 55.2 |

## C.3 EXAMINING INFERENCE TIME

Table 9 shows supplementary results of inference time for Table 1, Table 2, and Table 3. From **Top** of Table 9, VWFormer's inference time is faster than SegFormer. From **Bottom Left**, VWFormer's inference time is faster than UperNet. From **Bottom Right**, VWFormer's inference time is faster than FPN. Additionally, by comparing the results in the last row of **Bottom Left** and the first row of **Bottom Right**, it can be observed that VWFormer is faster than MaskFormer.

Table 9: **Top:** Frames/Sec.(FPS) of VWFormer and SegFormer. **Bottom Left:** FPS of VWFormer and UperNet. **Bottom Right:** FPS of VWFormer and MaskFormer. Evaluated on $512 \times 512$ crop for MiT and Swin-(Ti and S). Evaluated on $640 \times 640$ crop for ConvNeXt / Swin-(B, L, and XL)

| FPS ↑ | | MiT-B0 | MiT-B1 | MiT-B2 | MiT-B3 | MiT-B4 | MiT-B5 |
|-------|---|--------|--------|--------|--------|--------|--------|
| | SegFormer | 30.5 | 28.9 | 20.6 | 16.9 | 14.3 | 12.0 |
| | VWFormer | 30.8 | 29.4 | 21.1 | 17.4 | 14.6 | 12.5 |

| FPS ↑ | Swin-B | L | ConvNeXt-B | L | XL | FPS ↑ | Swin-Ti | S | B | L |
|-------|--------|-----|------------|------|------|-------|---------|------|------|-----|
| Uper. | 9.9 | 7.7 | 15.8 | 13.5 | 11.1 | Mask.-FPN | 22.1 | 19.6 | 11.6 | 7.9 |
| VW. | 12.0 | 9.0 | 19.7 | 17.2 | 14.6 | Mask.-VW | 23.1 | 20.7 | 11.9 | 7.9 |

## C.4 BREAKDOWN OF PERFORMANCE GAINS

Table 10 shows a breakdown of performance gains within Cityscapes which has 19-class segments. The upper results are obtained by MiT-B5 paired with SegFormer head (mIoU 82.26%) and the lower results are obtained by MiT-B5 paired with our VWFormer (mIoU 82.87%).

The bold number is the class that the counterpart performs better than ours. Except for the "truck" class where SegFormer outperforms ours largely, which seems like a biased result, on the 'wall', 'sky', and 'train' SegFormer only slightly outperforms ours (by avg. 0.2%). And on the other 15 classes, Ours shows consistent superiority to SegFormer (by avg. 1.4%).

Table 10: **Top:** Nine classes performance comparison of SegFormer and VWFormer on Cityscapes. **Bottom:** Ten classes performance comparison of SegFormer and VWFormer on Cityscapes.

| IoU | road | sidewalk | building | wall | fence | pole | light | sign | vegetation |
|-----|------|----------|----------|------|-------|------|-------|------|------------|
| Seg. | 98.5 | 87.3 | 93.7 | **68.6** | 65.7 | 69.5 | 75.6 | 81.8 | 93.2 |
| VW. | 98.5 | 87.6 | 94.0 | 68.4 | 68.7 | 73.0 | 77.3 | 84.5 | 93.5 |

| IoU | terrain | sky | person | rider | car | truck | bus | train | motorbike | bicycle |
|-----|---------|-----|--------|-------|-----|-------|-----|-------|-----------|---------|
| Seg. | 66.2 | **95.7** | 85.3 | 69.6 | 95.6 | **85.7** | 91.7 | **84.7** | 73.8 | 80.7 |
| VW. | 66.3 | 95.4 | 86.8 | 71.2 | 96.2 | 77.4 | 93.1 | 84.6 | 75.2 | 82.2 |

# D  SOME DETAILS

## D.1  DETAILS OF VWFORMER CAPACITY SETTING

Sec. 4 and Fig. 4 indicate the flow of channel numbers is 512(output of multi-layer aggregation)$\rightarrow$ 2048(concatenation of learnt multi-scale representations)$\rightarrow$ 512(output of multi-scale aggregation)$\rightarrow$ $512 + 48 = 560$(concatenation of LLE)$\rightarrow$ 256(final output of VWFormer).

For some lightweight backbones, such channel settings incur too much computational burden. We further introduce an efficient setting for VWFormer to cooperate with lightweight backbones such as SegFormer-B0 and SegFormer-B1. The new flow of channels is 128 (output of multi-layer aggregation)$\rightarrow$ 512(concatenation of learned multi-scale representations)$\rightarrow$ 128(output of multi-scale aggregation)$\rightarrow$ $128 + 32 = 160$(concatenation of LLE)$\rightarrow$ 128(final output of VWFormer)

## D.2  DETAILS OF DATASET

Cityscapes is an urban scene parsing dataset that contains $5,000$ fine-annotated images captured from 50 cities with 19 semantic classes. There are $2,975$ images divided into a training set, 500 images divided into a validation set, and $1,525$ images divided into a testing set.

ADE20K is a challenging dataset in scene parsing. It consists of a training set of $20,210$ images with 150 categories, a testing set of $3,352$ images, and a validation set of $2,000$ images.

COCOStuff-164K is a very challenging benchmark. It consists of 164k images with 171 semantic classes. The training set contains 118k images, the test-dev dataset contains 20K images and the validation set contains 5k images.

## D.3  DETAILS OF IMPLEMENTATION

Experiments of comparison with SegFormer 1 and UperNet 2 are implemented based on the MM-Segmentation codebase. In addition, ablation studies are done with MMSegmentation. Experiments comparing with MaskFormer and Mask2Former are implemented based on the Detectron2 codebase. The computing server on which all experiments are run has 16 Tesla V100 GPU cards. For other methods' results, we report the number shown in their papers.

# E  QUALITATIVE RESULTS

As shown in Fig. 8 and Fig. 9, we present more qualitative results on ADE20K of SegFormer and VWFormer with MiT-B5 as the backbone. The yellow dotted box focuses on the apparent visualization difference between them and the Ground Truth (GT). Compared to SegFormer's results, VWFormer improves the inner consistency of objects. Taking the bedroom (the first row shown in Fig 8 as an example, part of the shelf that is near the bed is misidentified as the shelf by Seg-Former, and the boundary between the bed and shelf is extremely unclear. In contrast, VWFormer segments the two objects very finely, which provides a coherent boundary. Moreover, we observe that with VWFormer similar objects are hardly confused. For example, in the living room shown in the last row of Fig. 9, SegFormer mistakes the sofa for an armchair. And in the first row of Fig. 9, SegFormer mistakes the blind for windowpanes. However, VWFormer accurately distinguishes between the sofa and the armchair, as well as between the blinds and the windowpanes.

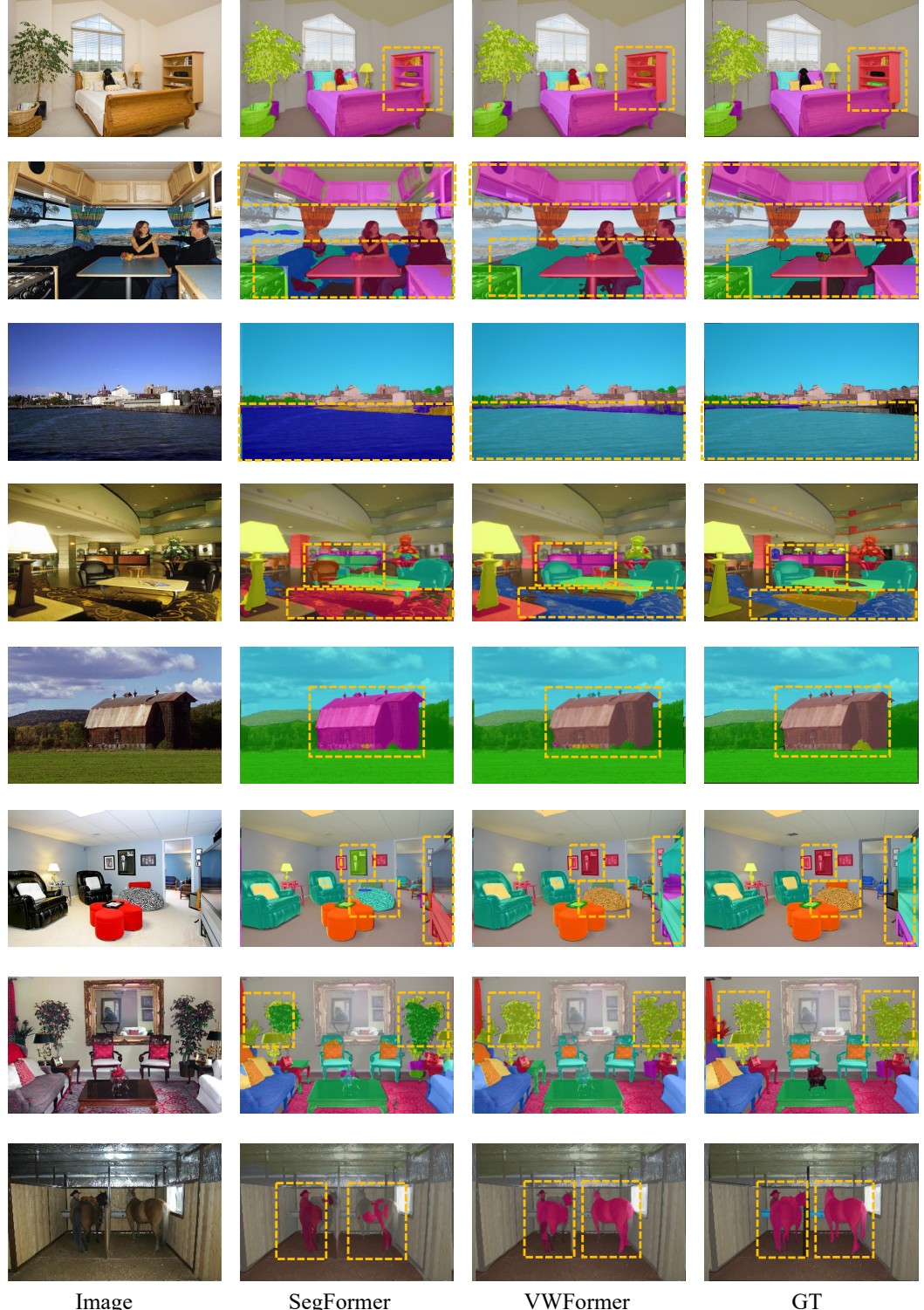

Figure 8: Qualitative results of ADE20K validation set. MiT-B5 serves as the backbone

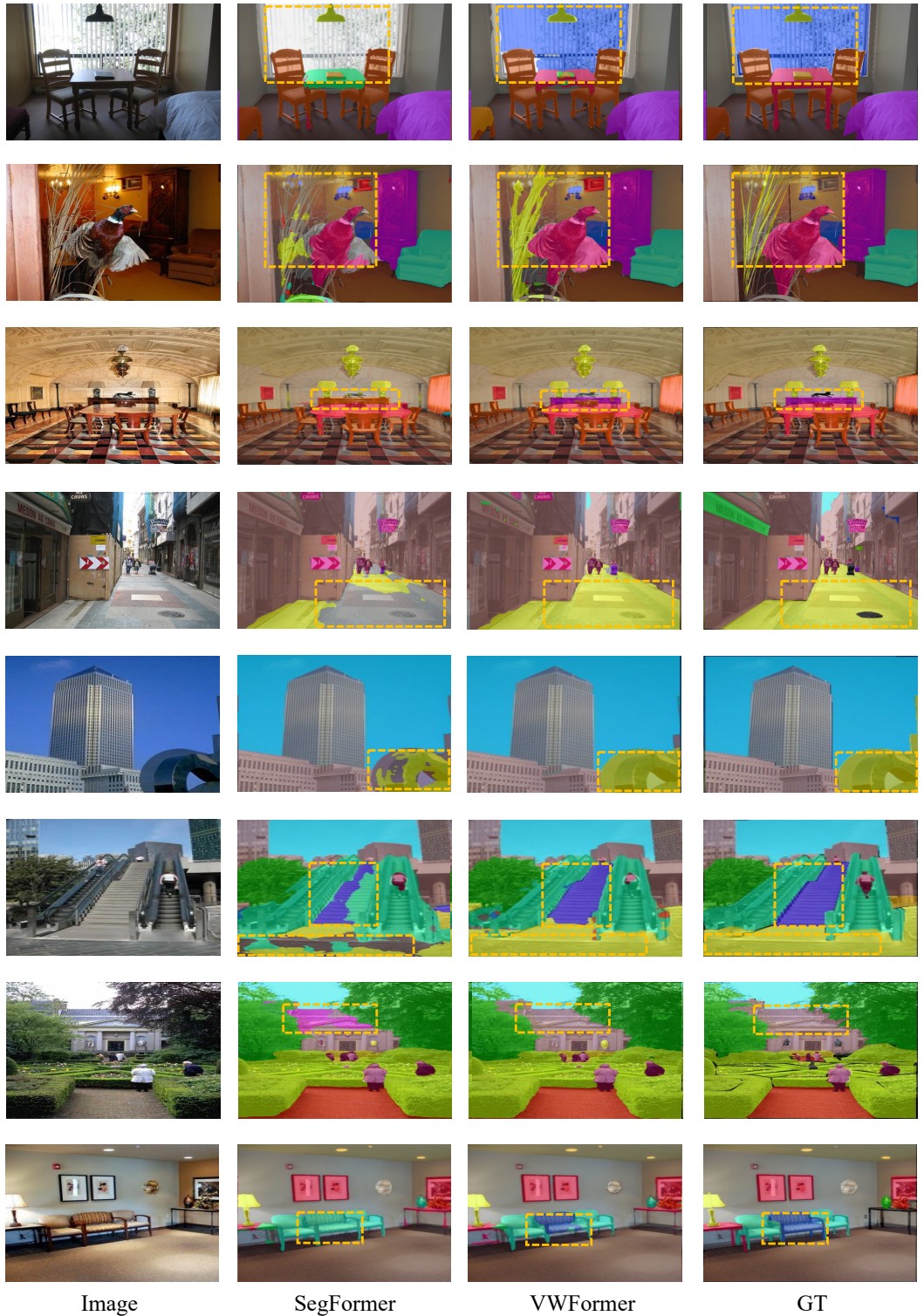

| Image | SegFormer | VWFormer | GT |

Figure 9: Qualitative results of ADE20K validation set. MiT-B5 serves as the backbone.

