# OpenReview forum: "Multi-Scale Representations by Varying Window Attention for Semantic Segmentation"
_ICLR.cc/2024/Conference — ICLR 2024 poster_

### Official Review · Reviewer_L266 · 2023-10-19

**Soundness:** 3 good
**Presentation:** 2 fair
**Contribution:** 2 fair
**Rating:** 6
**Confidence:** 5

**Summary:**

The paper delves into the topic of multi-scale representations and identifies two primary challenges: "scale inadequacy" and "field inactivation." The authors introduce the LWA model and, based on this, design the VWFormer as a decoder to enhance multi-scale representation capabilities. Through the effective LWA module, VWFormer outperforms most compute-friendly multi-scale decoders, such as FPN and MLP decoders.

**Strengths:**

Extensive experiments are conducted on Cityscapes, ADE20K, and COCOStuff-164K datasets. Despite reducing computational demands and the number of parameters, the proposed method demonstrates relative improvements compared to other approaches.

This work provides a thorough mathematical derivation of its method and showcases how to address the challenges of scale inadequacy and field inactivation.

This work presents a lot of ablation experiments to verify that the proposed methods outperform existing mechanisms.

The visualization results are interesting and provide many insights into multi-scale feature representations for semantic segmentation.

**Weaknesses:**

1. The manuscript requires structural refinement. Although the paper proposes a new methodology, the empirical validation predominantly resides in the appendix. The core content appears to be overly enmeshed in granular theoretical derivations, leading to a somewhat convoluted narrative structure.
2. The delineation between the figures, tables, and the main text is ambiguous. It is advisable to render the captions in a font slightly smaller than the primary text to enhance clarity. The layout of the illustrations also warrants improvement. For instance, in Figure 2, it would be beneficial to incorporate descriptions of specific letters either within the caption or designated sections of the image. The term "PE" is presented here, yet a precise elucidation is deferred until Section 3.3.
3. The visualization results do not substantiate the issues you've raised. The quantified outcomes displayed in Figure 5 merely illustrate the discriminative results of the proposed network in comparison to Segformer for a specific image region. If the method you presented exhibits a substantial improvement over Segformer, I believe it would be acceptable. However, given that the data only indicates approximately a 1% enhancement relative to Segformer. There is a lack of targeted visualization results addressing the issues of scale inadequacy and field inactivation.
4. The author introduced "varying window attention," but neglected a comparative analysis with other similar methods, such as the 'Shifted Window' in the Swin-Transformer.
5. Some state-of-the-art segmentation methods like HRViT, SegViT-v2, ViT-adapter, Lawin Transformer, etc. could be compared in the experiments to better verify the superiority of your proposed architecture.

Sincerely,

**Questions:**

1. Why not experiment with the same Encoder paired with different decoders (more than three)? This can help better show the consistent effectiveness of your approach.
2. The computational load shown in Equation (2) appears highly reminiscent of the Swin Transformer. Could you possibly incorporate a comparison with the Swin Transformer?
3. In Figure (3), you've employed an operation akin to that of Swin. However, after image concatenation in Swin, a mask is applied. As you concatenate features from different regions, how do you prevent interference or cross-effects between these diverse area features?
4. SegFormer utilizes an MLP Decoder, which doesn't encompass an attention mechanism. Yet, in your VWA mechanism, you've incorporated an attention mechanism. How does this lead to a reduction in parameter count? This can be better analyzed.
5. In the experimental section, could the results be organized based on the size of the modules in a systematic order?

Sincerely,

---

> ### Author Response · Authors · 2023-11-15
> **Thanks for your comments! Any new questions please let me know!**
>
> ### **For Weaknesses 1&2**:
> ----
> Thanks for the advice and careful review. We will refine the structure, delineation, and core content. We will check all captions, e.g. explaining PE in Fig. 2(b), and adjust the font size in all Figures to make readers clearly understand each of them. But with openness, I think it should be noted that two reviewers of all 4 think the presentation is good, one reviewer of all 4 thinks the presentation is fair, and two reviewers of all 4 think it is easy to follow.
>
> ### **For Questions 1&5**:
> ----
> > Experiment with the same Encoder paired with more than three decoders to show superiority consistently
>
> > Could the results be organized in an ascending order of module size ?
>
> Good idea to show our consistent superiority. The backbone choice depends on what the decoder's official paper used or what most papers paired it with. And the order in our paper is based on the performance of compared decoders.  Below is the result obtained by the same Encoder--Swin Transformer--paired with FPN, MLP-decoder, UPerNet, and our VW, respectively on ADE20K, and the order of other decoders is module-size ascending.
>
> | mIoU | Swin-Ti | Swin-S | Swin-B | Swin-L |
> | :----: | :----: | :----: | :----: | :----: |
> | FPN | 44.7 |  48.0 | 51.7 | 53.3 |
> | MLP | 46.2 | 48.4 | 52.5 | 53.9 |
> | Uper | 45.8 | 49.2 | 52.4 | 54.1 |
> | VW. | 46.9 | 50.4 | 53.6 | 55.4 |
>
> Likewise, we will rearrange the order of the compared modules in the paper according to their sizes.
>
> ### **For Weakness 3 and Question 4**:
> ----
> Your concern involves that Figure 5 can not substantiate the issues we have raised, i.e. scale inadequacy and field inactivation, and the improvement of about 1% is not substantial over segformer to show the effectiveness of addressing these issues. We will answer this concern from TWO perspectives.
>
> **(1)** I think visualization results showing the issues are not in Figure 5 but in Figure 1. In the beginning, the paper used EFR visualization to show the issue. It can be seen from Fig. 1(e) that the multi-scale feature of SegFormer only has the local and global scales(i.e. scale inadequacy) and some areas within the receptive field have almost zero values(i.e. field inactivation), as analyzed in Appendix A. The visualization of Figure 1 is averaged on many ADE20k val images, so I guess it might be too abstract to substantiate the proposed issue. THEREFORE, we sincerely request you to take a look at this [ANONYMOUS LINK](https://drive.google.com/file/d/1fKaeJBH8fFKDA4VM9X-ds4GTttfUdJ_E/view?usp=sharing) in which the specific ADE20K val image with EFRs of segformer and VWFormer are analyzed contrastively. We hope this new visualization will help to understand the receptive issue of SegFormer and show the strengths of VWFormer's multi-scale learning.
>
> **(2)** For comparison of VWFormer to segformer, meanIoU is not the only metric to be considered. From Table 1, you can see that besides meanIoU Table 1 reported params, flops, and memory usage. Why do we think the overall metrics comparison is meaningful? First, An improvement with no more computational budgets is a PARETO improvement, and especially considering that our motivation is to improve multi-scale representations of classical methods, the PARETO optimum is more favorable. Second, VWFormer is architecturally similar to adding VWA to the MLP-decoder so you naturally propose the **Question 4**. The answer can be very simple. Towards PARETO, We weakened the powerful multi-layer aggregation (MLA) module of VWFormer by reducing its original dimensionality (this is one step. Also along with LLE introduced in Sec.4.3 and a more efficient setting of VWFormer introduced in Appendix F) so our VW. can even be much more efficient than Seg. according to Table 1. At the beginning of this research, we ran VWFormer by keeping the original channel setting of MLA (768 or 2048), the improvements of VW. over Seg. become around 1.5% to 2.3%.
> |mIoU on ADE20K | B0 | B1 | B2 | B3 | B4 | B5 |
> | :- | :-: | :-: | :-: | :-: | :-: | :-: |
> |Seg. (original setting, shown in **Table 1**) | 38.0 | 43.1 | 47.5 | 50.0 | 51.1 | 51.8|
> |VW. (keeping original setting, **NOT** shown in Table 1) | 40.3 | 44.8 | 49.5 | 51.6 | 52.6 | 53.5|
> |VW. (with channel reduction and LLE, shown in **Table 1**) | 39.6 | 44.0 | 49.2 | 50.9 |  51.6 | 52.7|
>
> **To be continued.**

---

> ### Author Response · Authors · 2023-11-15
> **Thanks for your comments! Any new questions please let me know! (Cont.)**
>
> ### **For Weakness 4 and Questions 2&3**:
> ----
> > neglected comparative analysis to similar methods, e.g. Swin.
>
> This paper just begins by comparing similar methods including the Swin. For Swin, the analysis reveals that the 'Shifted-Window' mechanism directly incurs the field inactivation issue (please see **Introduction, Figure 1(d)** showing the shift-pattern of Swin's receptive field, and its captions cited **Appendix A**).
>
> > an operation akin to that of Swin
>
> After reading your questions 2&3, I believe a clearer distinction between VWA and Swin attention may be helpful : ) **a**. Swin is of Self-attention. VWA is of Cross-attention. **b**. Swin shifts every window's location and does nothing about its size. VWA fixes every window's location and varies (enlarges) its context window's size (so VWA needs no mask). **c**. If Swin increases the receptive field for multi-scale learning, it can only stack itself upon itself and finally will meet the field inactivation issue. But if VWA increases the receptive field, it simply varies the context window's size, which should be computationally expensive but its incorporated DOPE and pre-scaling will handle this.
>
> >computational load of Eq.(2) reminiscent of Swin
>
> The only similarity between Swin and VWA is they are both as computationally efficient as LWA. Question 2 also focuses on the computational load. To avoid misunderstandings because in Summary you said "the authors introduced LWA", it needs to be clarified that LWA computed by Eq. 2 is a preliminary idea, neither our contribution nor Swin's contribution. Whether it's the Swin or our proposed VWA, both are extensions of LWA. As explained in point **b** above, Swin only changes the location of the Local Window but does not change its size. Hence, Swin's computational load is the same as that of LWA, given by $4(HW)(C^2) + 2(HW)(P^2)C$ in Eq. 2.
>
> Tip: In the paper, we presented the computation of LWA straightforwardly. Since you asked this question, if you want more details about how $4(HW)(C^2) + 2(HW)(P^2)C$ is obtained we will show you here. If you already know it, just skip to <For Weakness 5>. Assuming the feature map is of shape ($H, W, C$) and it can be split uniformly into ($H/P \times W/P = \frac{HW}{P^2}$) patches with shapes of ($P \times P \times C$), for the LWA implements self-attention on one patch, its computation consists of two parts, first is mapping query, key, value, and output, so the first part of computation is ($4 \times P \times P \times C_{in} \times C_{out} = 4{P^2}{C^2}$). And second is attention computation including calculating attention maps and doing weighted summation, both of which are ($P^2 \times P^2 \times C = {P^4}C$) indicating that every pixel's feature of the patch will multiply itself elementwise and every pixel does weighted summation over all pixels according to the attention maps, so the second part of the computation is ($2 {P^4}C$). Now that for one patch the computation cost is $4{P^2}{C^2}+2{P^4}C$, for the whole feature map the computation cost is $[\frac{HW}{P^2} \times (4{P^2}{C^2}+2{P^4}{C})] = 4{(HW)}{(C^2)}+2{(HW)}{(P^2)}C$
>
> ### **For Weakness 5**:
> ----
> Experiments in the paper were conducted mostly to show that VWFormer is the best MSD for semantic segmentation.
>
>
> >HRViT-b1,b2, and b3
>
> HRViT takes the MLP-decoder as the default MSD. If HRViT is paired with VWFormer, the improvements are reported as follows:
> | mIoU on ADE20K | HRViT-b1 | HRViT-b2 | HRViT-b3 |
> | :- | :-: | :-: | :-: |
> |HRViT-MLP | 45.6 | 48.8 | 50.2|
> | HRViT-VW. | 46.9 | 50.4 | 51.6|
>
> >SegViT-V2
>
> SegViT-V2 is a decoder designed for ViT. We evaluate VW. paired with BEiTv2.
>
> | mIoU on ADE20K | BEiT-V2-L  |
> | :- | :-: |
> | SegViT-V2 |  58.2 |
> | VW. | 59.0 |
>
> >ViT-Adapter
>
> Adapter is equipped to ViT paired with UperNet as MSD. The UperNet is replaced with VWFormer and we get the following improvements:
> | mIoU on ADE20K | ViT-B | ViT-L |
> | :- | :-: | :-: |
> |Adapter-Uper. | 52.5 | 54.4|
> |Adapter-VW. | 54.1 | 55.8 |
>
>
> > Lawin Transformer
>
> Indeed, VW. is Version 2 (or called thoroughly improved version) of Lawin.

---

> > ### Comment · Reviewer_L266 · 2023-11-16
> > **Comment**
> >
> > The reviewer would like to thank the authors for the responses and the added analyses. The added results show that the proposed method can help improve the segmentation performance of recent methods like HRViT.
> >
> > The response helps to address many concerns and the reviewer would like to improve the rating. Please incorporate these analyses in the final version.

---

> > > ### Author Response · Authors · 2023-11-17
> > > **Thanks for improving the rating!**
> > >
> > > It is a pleasure to know that our responses can address your concerns. We are making the paper incorporate these analyses.

---

### Official Review · Reviewer_Vryj · 2023-11-01

**Soundness:** 3 good
**Presentation:** 2 fair
**Contribution:** 2 fair
**Rating:** 5
**Confidence:** 3

**Summary:**

This paper points out two problems existing in the previous multi-scale semantic segmentation representation: scale inadequacy and field inactivation. To solve these two problems, this paper presents its method - VWA (varying window attention). It begins by splitting local window attention (LWA) into query window and context window, which are variable so that queries learn representations on a specific scale. Then, to reduce memory consumption, the pre-scaling principle, densely overlapping patch embedding(DOPE), and copy-shift padding mode are proposed. Finally, VWFormer is devised for semantic segmentation, which shows its effectiveness in terms of performance and reducing cost.

**Strengths:**

+ This paper makes full use of multi-scale features for fusion and proposes VMA to obtain multi-scale representation.
+ The experiments and visual results show the problems that existed in the previous work, namely, scale inadequacy and field inactivation, which is insight.
+ The VMFormer frame is more clearly drawn and is easy to follow.

**Weaknesses:**

- Too much space is spent to explain the memory usage problem, resulting in some experimental results, such as Table 9 and some ablation experiments, which can only be placed in the appendix.
- Lack of ablation experiments on the copy-shift padding mode. It is mentioned in the paper that filling with zero will cause attention collapse, but there are no experimental results to explain it.
- There is no comparison between VWA and LWA.
- The pre-scaling principle is to reduce memory usage, and the impact of pre-scaling on performance is not reflected.
- The method part is not abundant, such as the short path branch. Overall, it seems that the original is not high, mainly VWA.
- Inference time is not reflected in the experiments.

**Questions:**

(1) For copy-shift padding mode(CSP), why not take the adjacent part to padding?
(2) What is the short path branch?
(3) What does pre. or post. in Table 7 refer to?  PE refers to padding with zero or CSP?

---

> ### Author Response · Authors · 2023-11-15
> **Thanks for your comments! Any new questions please let me know!**
>
> ### **For Weakness 1**:
> ---
> Thanks for the structure advice. We will move key ablation studies to the main text and some equations explaining memory usage to the appendix, re-balancing the math and experiments.
>
> ### **For Weakness 2 and Question 1**:
> ---
> > Lack of ablation experiments on the copy-shift padding mode
>
> The ablation experiment on the copy-shift padding mode is **Table 8 Left** in the paper. It will be moved to the main text.
>
> > Why not take the adjacent part to padding
>
> I think you refer to adjacent part padding as the symmetric padding mode in Pytorch (If I mistake it please tell me). In this research, we have tried many techniques to tackle attention collapse, the symmetric padding mode has also been a candidate but it did not provide high improvements like CSP. This is because the symmetric mode only copies the already existing feature within the context window but CSP offers the window at the edge and corner a wider contextual receptivity.
>
> What does the "wider contextual receptivity" indicate? For example, given a feature map of $64\times64$, the local window is $8\times8$, and the context window is $32\times32$. If the VWA is at the upper left corner, the context window's coverage on the feature map is the feature map's [-12:20, -12:20] (here minus means outside feature map), so $12\times20+20\times12 +12\times12$ paddings are needed for the context window. That means, with symmetric padding mode, the padding elements are shifted from [0:12, 0:20], [0:20, 0:12], and [0:12, 0:12] of the feature map, all of which are already covered by the context window. But if using CSP, from Eq. 17-18, the padding elements are shifted from [20:32, 0:20], [0:20, 20:32], and [20:32, 20:32], and now the context window's coverage is changing and not initial [-12:20, -12:20] of the feature map but becomes [0:32, 0:32] of the feature map.
>
> Below is our previous trial result on the adjacent padding. It can be seen the adjacent padding's improvement is not that high compared to copy&shift.
> | padding mode | mIoU(/MS) |
> | :----: | :----: |
> | zero padd. | 51.6 / 52.6 |
> | adjacent |  51.9 / 52.9 |
> |copy&shift |  52.3 / 53.6|
>
> ### **For Weakness 3**:
> ---
> Sec 3.2 and Fig. 2 show how LWA becomes VWA and conclude that LWA is a special VWA with $R=1$. Table 6 analyzes the impact of LWA on VWFormer. If you want some particular comparison, please let me know and I will show them soon : )
>
> ### **For Weakness 4 and Question 3**:
> ---
> >  The impact of pre-scaling on performance is not reflected & What does pre. or post. in Table 7 refer to?
>
> The impact of pre-scaling is reflected in Table 7 because pre. or post. refers to Pre-scaling and Post-scaling.
>
> Sec. 3.2 shows that VWA with **no scaling** leads to unaffordable computation cost and memory usage. Sec. 3.3 shows that  **post-scaling** is a naive solution able to remove the extra computation but unable to handle the memory usage, and **pre-scaling** is proposed for handling both of computation cost and memory usage. Table 7 compares these three strategies. Then you question about:
>
> > PE refers to padding with zero or CSP?
>
> PE refers to patch embedding, just the same as 'patch embedding' in ViT[1]. Regardless of the post-scaling or pre-scaling, there must be specific scaling operators. PE is the operator doing post-scaling. For pre-scaling, we propose DOPE paired with PE, specifically DOPE--> PE, to implement it, as introduced in Sec 3.3.
>
> ### **For Weakness 5 and Question 2**:
> ---
> > What is a short path branch
>
> A short path (means with no heavy components along this path) branch refers to a residual connection[2] or an identity mapping i.e. 1x1 conv[3].
>
> > The original is not high, mainly VMA
>
> The basic idea of VMA is original. The architecture of VWFormer, even if it has no VWA, is original. The DOPE with pre-scaling strategy is original because it is a new way to embed patches for a Transformer-like vision framework. The copy&shift padding is original because it is a new padding mode different from off-the-shelf padding modes existing in Numpy or Pytorch.
>
>
>
> Ref.
>
> -- [1] An image is worth 16x16 words: Transformers for image recognition at scale. In ICLR, 2022
>
> -- [2] Deep residual learning for image recognition. In CVPR, 2016.
>
> -- [3] Identity mappings in deep residual networks. In ECCV, 2016
>
> **To be continued.**

---

> ### Author Response · Authors · 2023-11-15
> **Thanks for your comments! Any new questions please let me know! (Cont.)**
>
> ### **For Weakness 6:**
> ---
> > Inference time
>
> To compare the inference time, we supplement Tables 1, 2, and 3 in the paper by assessing the frame per second (FPS), respectively.
>
> Comparison of Ours to SegFormer
> | FPS | MiTB0 | MiTB1| MiTB2| MiTB3 | MiTB4| MiTB5|
> | :----: | :----: | :----: | :----: | :----: | :----: | :----: |
> | Seg. | 50.5 | 46.2 | 30.9 | 22.1 | 15.5 | 11.9 |
> | VW. | 53.4 | 49.8 | 35.3 | 26.8 | 19.2 | 14.0 |
>
> Comparison of Ours to Swin-Uper.
> | FPS | Swin-Ti | Swin-S | Swin-B | Swin-L |
> | :----: | :----: | :----: | :----: | :----: |
> | Uper. | 18.5 | 15.2 | 8.7 | 6.2 |
> | VW. | 24.4 | 22.0 | 15.9 | 8.8 |
>
> Comparison of Ours to Swin-MaskFormer
> | FPS | Swin-Ti | Swin-S | Swin-B | Swin-L |
> | :----: | :----: | :----: | :----: | :----: |
> | Mask.-FPN | 22.1 | 19.6 | 12.6 | 7.9 |
> | Mask.-VW. | 23.1 | 20.7 | 12.9 | 7.9 |
>
> In summary, our inference time is faster than SegFormer and MaskFormer, much faster than UperNet, and slightly faster than FPN.

---

> > ### Comment · Reviewer_Vryj · 2023-11-22
> >
> > The authors have solved my concerns. I would like to improve the rating. I suggest that the authors had better present the new analyses and results in the final version.

---

> ### Author Response · Authors · 2023-11-21
> **The discussion period will end in two days！**
>
> Dear Reviewer Vryj,
>
> May I inquire if you have received our responses to your comments on the paper (submission ID:104)?
>
> ALSO, it might be worth considering that the most confident reviewer (L266) has improved the rating so currently there are two most confident reviewers (L266 and u8Vk) suggesting the paper is above the acceptance threshold.
>
> At the moment the discussion period concludes, we will update a new version of the paper with additional analyses inspired by your comments.
>
> Sincerely,
>
> Authors of Paper (submission ID 104)

---

> ### Author Response · Authors · 2023-11-23
> **Thanks for the improvement! We look forward to your new rating!**
>
> It is a pleasure to know that we can solve your concerns. We are incorporating new analyses and results into the paper.

---

### Official Review · Reviewer_u8Vk · 2023-11-02

**Soundness:** 3 good
**Presentation:** 3 good
**Contribution:** 3 good
**Rating:** 6
**Confidence:** 4

**Summary:**

This paper observes the limitation of local window attention and proposes an efficient modification to enlarge the receptive field while keeping the computation efficiency. Specifically, it designs a pre-scaling strategy, which cleverly reduces the extra computation of enlarging window size. This paper presents detailed computation comparisons with existing methods and verifies the effectiveness of the proposed methods on several benchmarks.

**Strengths:**

1. The paper starts from the visual observations, then goes into detailed computation analysis, and further proposes modifications to enhance. The logic between the whole paper is very smooth and easy to follow.

2. The detailed complexity analysis substance the efficiency of the proposed method. Besides, the experimental results further verify computation efficiency compared to existing methods.

3. Quantitative analysis verifies the effectiveness of the proposed methods, highlighting the significance of varying window sizes.

**Weaknesses:**

1. The proposed copy-shift padding is a bit weird. It mimics the rolling operation in numpy and torch. I wonder why adopting this instead of symmetric or constant padding.

2. Fig 2b and Sec 3.2 are not well-aligned. In Fig 2b there's a PE operation, which doesn't occur in Sec 3.2, which can confuse the readers.

**Questions:**

1. Are there any ablation studies indicating the effectiveness of copy-shift padding?

2. From Table 7 in Supp, it seems avg_pool can be a substitution of PE, then I wonder what the performance will be like if the rescaling method "DOPE --> avg_pool --> 1x1 conv" is taken.

---

> ### Author Response · Authors · 2023-11-15
> **Thanks for your comments! Any new questions please let me know!**
>
> ### **For Weakness 1 and Question 1**:
> ---
> > Copy&shift mimics the rolling operation
>
> Very insightful opinions. CSP seems like the roll function but very very vdifferent from it. If only seeing Figure 3, CSP is like "copy&shift" and the roll function is like "cut&shift". HOWEVER, more significantly, the roll function can only operate the elements of the feature map's edge parts, cutting an edge and shifting it to another edge. But from Eq. 17 and 18, CSP is proposed for copying the elements that are nearby outside the context window's inside-feature edges and then shifting these elements to padding so that the context window can cover them instead of zeros when the context window is at the edge or corner, which is the roll function in Numpy or Pytorch can not achieve.
>
> > Any ablation studies indicating the effectiveness
>
> At the bottom of **page 15**, **Table 8 Left**, shows the effectiveness of CSP and that the zero padding mode impairs performance largely.
>
> > Symmetric or constant padding.
>
> In this research, we have tried many techniques to tackle attention collapse, and symmetric padding mode has also been a candidate but it did not provide high improvements like CSP. We think this is because the symmetric mode only copies the already existing feature within the context window but CSP offers the window at the edge or corner a wider contextual receptivity [1]. For constant padding, zero padding is a kind of constant padding so we are not very clear on what constants may be useful for this problem and all candidates having been tried by us are dynamic methods.
> | padding mode | mIoU(/MS) |
> | :----: | :----: |
> | zero padd. | 51.6 / 52.6 |
> | symmetric |  51.9 / 53.0 |
> |copy&shift |  52.3 / 53.6|
>
>
>
> ### **For Weakness 2**:
> ---
> We will revise the text-figure alignment issue in the paper to make readers clear. And we will polish every caption to let readers know what the used term in every Figure means : )
>
> ### **For Question 2**:
> ---
> Very good question, you must have carefully read the paper. Although avg_pool can be a substitution for PE, the 2nd and 3rd rows in Table 7 show that avg_pool leads to an unignorable performance drop (53.7%-->52.8%). So if using avg_pool-->conv1x1 to substitute PE in the "DOPE-->PE" operation, there will be an unavoidable decrease in performance. From another view of efficiency, the 2nd and 3rd rows in Table 7 show that avg_pool--> conv1x1 is much more efficient than PE. But that works with the post-scaling strategy. With the pre-scaling strategy, according to the analysis in Sec. 3.3 and Eq. 13-15, it can be seen that PE is the same as a linear mapping. So PE in the "DOPE-->PE" operation with a pre-scaling strategy is efficient enough and it is redundant to replace it with avg_pool --> conv1x1 for more efficiency. Below is the comparison of "DOPE-->PE" with "DOPE-->avg_pool-->conv1x1", showing the performance drop caused by the substitution.
> | rescaling method | mIoU(/MS) |
> | :---- | :----: |
> | DOPE --> PE | 52.3 / 53.6 |
> | DOPE --> avg_pool-->conv1x1 |  51.8 / 52.8 |
>
> Tip:
>
> [1] What does the "wider contextual receptivity" indicate? For example, given a feature map of $64\times64$, the local window is $8\times8$, and the context window is $32\times32$. If the VWA is at the upper left corner, the context window's coverage on the feature map is the feature map's [-12:20, -12:20] (here minus means outside feature map), so $12\times20+20\times12 +12\times12$ paddings are needed for the context window. That means, with symmetric padding mode, the padding elements are copied from [0:12, 0:20], [0:20, 0:12], and [0:12, 0:12] of the feature map, all of which are already covered by the context window. But if using CSP, from Eq. 17-18, the padding elements are copied from [20:32, 0:20], [0:20, 20:32], and [20:32, 20:32], and now the context window's coverage is not initial [-12:20, -12:20] of the feature map but becomes [0:32, 0:32] of the feature map.

---

> > ### Comment · Reviewer_u8Vk · 2023-11-23
> > **Issues well solved**
> >
> > All my issues have been solved and I will keep the score

---

> ### Author Response · Authors · 2023-11-22
> **The discussion period will end in less than two days!**
>
> Dear Reviewer u8Vk,
>
> May I inquire if you have received our responses to your comments on the paper (submission ID:104)?
>
> ALSO, it might be worth considering that the most confident reviewer (L266) has improved the rating (rating 6 with confidence 5). Would you like to adjust your rating or confidence?
>
> Sincerely,
>
> Authors of Paper (submission ID 104)

---

### Official Review · Reviewer_1jxp · 2023-11-07

**Soundness:** 3 good
**Presentation:** 3 good
**Contribution:** 3 good
**Rating:** 6
**Confidence:** 3

**Summary:**

The submission introduces Varying Window Attention (VWA) to address scale inadequacy and field inactivation in semantic segmentation. VWA modifies the local window attention mechanism to dynamically adjust the receptive field, aiming to improve multi-scale representation learning. The paper also proposes a new multi-scale decoder, VWFormer, which incorporates VWA and demonstrates performance and efficiency gains on standard datasets. The work claims three main contributions: the VWA mechanism, the VWFormer decoder, and empirical improvements over state-of-the-art methods. Extensive experiments validate the effectiveness of the proposed methods.

**Strengths:**

+ Introduction of Varying Window Attention (VWA) to dynamically adjust receptive fields, addressing scale inadequacy and field inactivation.
+ Development of new principles such as pre-scaling, densely overlapping patch embedding (DOPE), and copy-shift padding mode (CSP) to enhance efficiency in receptive field variation.
+ Empirical improvements in mean Intersection over Union (mIoU) and reductions in computational cost (FLOPs).

**Weaknesses:**

- The paper may not sufficiently differentiate VWA from existing local window attention mechanisms.
- Potential concerns about the scalability and generalizability of the proposed method to different architectures or datasets.

**Questions:**

1. How does VWA fundamentally differ from existing attention mechanisms in handling multi-scale representations?
2. Include a breakdown of performance gains across different classes or segments within the datasets used.

---

> ### Author Response · Authors · 2023-11-15
> **Thanks for your comments! Any new questions please let me know!**
>
> ### **Above All**
> ---
> We noticed that you gave the score **Good** for all of Soundness, Presentation, and Contribution, yet the overall rating is marginally **below** the acceptance threshold. We are wondering if there might be a typo in the overall rating : )
>
>
> ### **For Weakness 1 and Question 1**:
> ---
> > How does VWA DIFFER from existing window attention mechanisms in handling multi-scale representation learning
>
> In the paper we called them 'multi-scale attention' introduced in the last line of Sec. 2.1. Please see the second paragraph of **Sec. 2.1, Fig. 6, Appendix B, and Table 9** (both ISANet and ANN in Table 9 are 'multi-scale attention').
>
> I am glad to present a summary of these related methods here (If you have some particular methods for comparing with our VWA, please let me know and I will compare them. To my knowledge, the listed works in the paper are all existing ones). ISANet[1] and GG-Transformer[2] have the same core idea but apply it to different tasks, semantic segmentation and image recognition, respectively. This idea first splits the image into windows to learn local representations and then interlaces the pixels among these windows so (pixels in) every window can learn global representations. The biggest problem with the ISANet-like method is they did not provide a possibility to learn representations of other scales, as illustrated in Fig. 6 (a). Table 9 compares ISANet with ours, showing that Our VWA outperforms them by ISANet margins.
>
> ANN[3], Focal Transformer[4], and Shunted Transformer[5] share the same core idea but apply it to different tasks. ANN is for semantic segmentation, and both Focal and Shunted are for image recognition. This idea depends on pooling filters to learn multi-scale representations (Appendix A analyzes the pooling's issue on multi-scale learning). One may question the proposed VWA also applies DOPE to downsample the image like pooling. The key point lies in that VWA changes the context window size so pixels in windows have varying receptive fields. However, the ANN-like method does not deal with the context window size so its receptive field is indeed fixed, as illustrated in Fig. 6(b). Table 9 also shows that VWA outperforms ANN by large margins.
>
> In short word, existing multi-scale attention, whether based on ISANet or ANN, confines the representation learning within the window. Only VWA liberates the representation learning beyond the window, covering any region.
>
> Ref.
>
> -- [1] Interlaced sparse self-attention for semantic segmentation. In IJCV, 2021
>
> -- [2] Glance-and-gaze
> vision transformer. In NIPS, 2022
>
> -- [3] Asymmetric non-local neural
> networks for semantic segmentation. In CVPR, 2019
>
> -- [4] Focal self-attention for local-global interactions in vision transformers. In NIPS, 2022
>
> -- [5] Shunted self-attention
> via multi-scale token aggregation. In CVPR, 2022
>
> **To be continued.**

---

> ### Author Response · Authors · 2023-11-15
> **Thanks for your comments! Any new questions please let me know! (Cont.)**
>
> ### **For Weakness 2 and Question 2**:
> ---
> >Scalability or Generalizability
>
> Good points, we will conduct in-depth research on them.
>
> BUT now, some good indicators can be presented. Please see Appendix F. It introduces that our method has two settings of different model sizes to match backbones with different model capacities, including the efficient setting specially designed for lightweight backbones like SegFormer-B0 and B1 compared with ours in Table 1, and a normal setting for regular backbones. We have also experimented with scalability on the MLA module of VWFormer, for example, scaling the channel from 512 to 2048. With Mask2Former and Swin-L, our method can improve mIoU from 58.3% to 58.8%. To show generalizability, the proposed method has been evaluated with MiT (B0-->B5), Swin Transformer (Ti-->L), MaskFormer, Mask2Former, Deformable Attention, and ResNet(50-->101) in the paper. The benchmarks contain three challenging datasets as introduced in Appendix G.
>
> > A breakdown of performance gains
>
> I will show a breakdown of performance gains within Cityscapes which has 19-class segments. The upper results are obtained by MiT-B5 paired with SegFormer head (mIoU 82.26%) and the lower results are obtained by MiT-B5 paired with our VWFormer (mIoU 82.87%).
>
> |method | road | side. | build. | wall | fence | pole | light | sign | vege. | terrain | sky | man | rider | car | truck | bus | train | motor | bike|
> | :-| :-: | :-: | :-: | :-: | :-: | :-: | :-: | :-: | :-: | :-: | :-: | :-: | :-: | :-: | :-: | :-: | :-: | :-: | :-: |
> |Seg. | 98.5 | 87.3 | 93.7 | **68.6** | 65.7 | 69.5 | 75.6 | 81.8 | 93.2 | 66.2 | **95.7** | 85.3 | 69.6 | 95.6 | **85.7** | 91.7 | **84.7** | 73.8 | 80.7|
> |VW. |98.5 | 87.6 | 94.0 | 68.4 | 68.7 | 73.0 | 77.3 | 84.5 | 93.5 | 66.3 | 95.4 | 86.8 | 71.2 | 96.2 | 77.4 | 93.1 | 84.6 | 75.2 | 82.2|
>
> The bold number is the class that the counterpart performs better than ours. Except for the "truck" class where SegFormer outperforms ours largely, which seems like a biased result, on the 'wall', 'sky', and 'train' SegFormer only slightly outperforms ours (by avg. 0.2%). And on the other 15 classes, Ours shows consistent superiority to SegFormer (by avg. 1.4%).
>
> Besides, we did a breakdown of performance gains within ADE20K which has 150-class segments. Since it has too many classes, I will list some important statistics. The comparable results are obtained from Swin-L-Mask2Former and Swin-L-Mask2Former with VWFormer in place of Deform-FPN. (1) The pure Mask2Former achieves 56.96% mIoU and ours achieves 58.27% mIoU. (2) In all 150 classes, their methods outperform ours in 58 classes, while ours outperform theirs in 92 classes. (3) In the 58 categories where their methods outperform ours, the average improvement is -4.7%. In the 92 categories where ours outperforms theirs, the average improvement is 5.2%.

---

> ### Author Response · Authors · 2023-11-21
> **The discussion period will end in two days！**
>
> Dear Reviewer 1jxp,
>
>
> May I inquire if you have received our responses to your comments on the paper (submission ID:104)?
>
> ALSO, it might be worth considering that the most confident reviewer (L266) has improved the rating so currently there are two most confident reviewers (L266 and u8Vk) suggesting the paper is above the acceptance threshold. Again, reviewer Vryj also expressed an intention to improve the rating.
>
> At the moment the discussion period concludes, we will update a new version of the paper with additional analyses inspired by your comments.
>
> Sincerely,
>
> Authors of Paper (submission ID 104)

---

> > ### Comment · Reviewer_1jxp · 2023-12-05
> >
> > Thanks for the detailed expaination. My initial concerns have been solved so I will raise my the rating.

---

### Author Response · Authors · 2023-11-23
**Author's explanation for the rebuttal revision.**

Thanks to constructive discussions with reviewers, the paper has undergone further improvements.

As the discussion period allows for updates before its conclusion, we have just uploaded a revised version of the paper. The key modifications are as follows:

* Unnecessary mathematical formulas, particularly in Section 4, and less important visualizations of attention maps have been removed (originally in Appendix Section C).

* Section 5.2.4, initially present in the main text, has been relocated to Appendix C.2 for better structure.

* The more crucial ablation studies, initially in Appendix D, have been moved to the main text Section 5.3 to address concerns raised by reviewer u8Vk regarding the accessibility of key experimental results.

* Following discussions with reviewer L266, we conducted ERF visualization analyses on specific images. These analyses, crucial for substantiating the issues we addressed, are now incorporated into Section 6 of the main text.

* Following discussions with reviewer L266, we compared our method with extra state-of-the-art methods. The results are presented in Appendix C.3.

* Discussions with reviewer Vryj led us to examine the inference time of our method. The results are now presented in Appendix C.4.

* Following discussions with reviewer 1jxp, we conducted a breakdown of performance gains. The results are included in Appendix C.5.

* We reviewed image captions to ensure that readers can understand the terms mentioned in the images.

* We improved the alignment between images and text to make readers quickly find relevant content.

* A thorough check for typos has been conducted, and redundant narratives have been removed for clarity.

These revisions aim to enhance the paper's clarity, readability, and scientific rigor.

At last, we appreciate the careful reviews and insightful comments from all reviewers.

Sincerely,

Authors of Paper (submission ID 104)

---

### Meta-Review · Area_Chair_ykhN · 2023-12-24

**Metareview:**

Varying Window Attention ("VWA") extends local (or "windowed") attention by cross-attending between a local window and larger and varying context windows. The enlargement of the context window is parameterized by the rate R, an architectural hyperparameter, where R = 1 reduces to standard local attention and R>1 multiplies the size of the context (Sec. 3.2 and Fig. 2). While direct computation of enlarged context windows would require additional time and memory, this can be controlled by spatial downsampling and channel reduction according to the same factor R (Sec. 3.3), which is implemented as a kind of patch embedding ("DOPE"). Along with DOPE a variant of padding ("CSP") is proposed to avoid degenerate attention weights at edges and corners, which would otherwise be worsened by the larger context patches. Finally, the proposed VWA is incorporated into a hierarchical/multi-scale architecture: the VWFormer. Experiments cover the standard datasets of Cityscapes and ADE20k (+ the bonus dataset of COCO-Stuff) and compare against popular and strong baselines (SegFormer, UperNet, MaskFormer, Mask2Former) and show small but consistent gains (0.5-1 point absolute) in accuracy (mIoU) without increased memory or operations (and sometimes even less). Finally, evaluating the proposed VWA against alternatives for increasing receptive field size (adaptive pooling, dilated convolution) demonstrates improved accuracy for both ResNet and Swin backbones. All-in-all the empirical results support the proposed extension of local attention: it achieves better accuracy for semantic segmentation with competitive or better computational efficiency.

Four experienced reviewers with expertise on segmentation, attention, and multi-scale/contextual processing rate the submission as borderline with three borderline accept (1jxp, u8Vk, L266) and one borderline reject (Vryj). Note however that Vryj [commented](https://openreview.net/forum?id=lAhWGOkpSR&noteId=VIqOdeph27) that the authors have solved their concerns and their rating would be improved, so although their rating remains a borderline reject, the AC has interpreted it as a borderline accept. Three reviewers raised their scores (1jxp, L266, Vryj) given the author response and author-reviewer discussion, with 1jxp confirming their improved rating during the reviewer-AC discussion. The authors incorporated a number of improvements prompted by reviewers ([summary](https://openreview.net/forum?id=lAhWGOkpSR&noteId=GgxHAmVUk8)) which can be verified by inspection of the revision. The authors also provided a confidential comment, to summarize the discussion, which the AC acknowledges. Additional comments following the author-reviewer discussion phase confirmed the final ratings.

The AC sides with acceptance. Although the reviewers are borderline, they are all positive, and the work contributes a clearly-explained variant of windowed attention and experimentally establishes its value for semantic segmentation on standard datasets and current baselines. This should be informative to the community on this topic itself, but also for extensions to other pixel-wise tasks, as has often been the case now given the convergence in architecture across vision tasks (whether convolutional or attentional).

To encourage the authors to polish their work, the AC offers the following miscellaneous feedback:

- Consider adding work on local/contextual processing and multi-scale aggregation to the related work, such as [​​Context Contrasted Feature and Gated Multi-scale Aggregation for Scene Segmentation](https://openaccess.thecvf.com/content_cvpr_2018/papers/Ding_Context_Contrasted_Feature_CVPR_2018_paper.pdf) (Ding et al. CVPR 2018).
- Consider reorganizing the text so that the conclusion is included in the main paper and not shunted into an appendix.
Please carefully edit the submission for proofreading, as even the name of the method has a typo (see Sec. 3: "Varing Window Attention").

**Justification For Why Not Higher Score:**

- The experiments are thorough within the scope of semantic segmentation, but only cover semantic segmentation. For a learning conference like ICLR it would be better still to show results across multiple pixel-wise vision tasks, or even a larger variety of vision tasks such as segmentation and detection.
- There are some organizational issues with the current edition of the work and there is a degree of reliance on the appendices. Reviewers have commented on this (L266 addresses this clearly and politely)  and the authors are encouraged to handle this in their revisions. Nevertheless the work must be evaluated as it is.

**Justification For Why Not Lower Score:**

- The proposed VWA is demonstrated to be effective and efficient. The experiments evaluate VWA and VWFormer on standard datasets against current baselines and further analysis examines VWA vs. alternative operations for increasing effective receptive field size.
- The paper is well-reasoned and justified in its progression from observations, through its methods, and finally the results. Reviewers agree on the clarity of the operations and architecture, and any clarifications requested have been made during the author response.

---

### Decision · Program_Chairs · 2024-01-16

Accept (poster)